# Simulated hydrological effects of grooming and snowmaking in a ski resort on the local water balance

Samuel Morin[1], Hugues François[2], Marion Réveillet[1,3], Eric Sauquet[4], Louise Crochemore[3,4], Flora Branger[4], Étienne Leblois[4], and Marie Dumont[1]

[1]Univ. Grenoble Alpes, Université de Toulouse, Météo-France, CNRS, CNRM, Centre d'Etudes de la Neige, Grenoble and Toulouse, France
[2]Univ. Grenoble Alpes, INRAE, LESSEM, Grenoble, France
[3]Univ. Grenoble Alpes, CNRS, IRD, IGE, Grenoble, France
[4]INRAE, RiverLy, Lyon, France

**Correspondence:** Samuel Morin (samuel.morin@meteo.fr)

**Abstract.** The presence of a ski resort modifies the snow cover at the local scale, due to snow management practices on ski pistes, especially grooming and snowmaking. Snow management exerts two-fold effects on the local hydrological cycle, through (i) abstraction and transfer of water used for snowmaking, and (ii) changes in water runoff due to added snow mass through snowmaking and/or delayed melting of the snowpack due to snow grooming. This induces a local pressure on water resources, which has seldom been addressed in scientific studies hitherto.

Here we introduce a method to compute the hydrological effects of snow management on ski pistes, and we apply and illustrate its results for the case study of the La Plagne ski resort in the Northern French Alps. The approach mainly relies on snow cover modelling using the Crocus snow cover driven by SAFRAN reanalysis and climate projections. Model results are evaluated against in-situ hydrological observations and show that the modelling approach, although very simplified for many hydrological processes, provides relevant information and insights in terms of the influence of snow-related processes on water resources.

Our study shows a visible impact of grooming, virtually eliminating snowmelt in winter, thus delaying the onset of snowmelt. While about 10% of the water used for snowmaking is estimated to be lost by evaporation through the ice formation process from the liquid water droplets, we find that, in the case studied, the annual scale alteration of water resources is limited, and estimated to be on the order of 1 to 2 %. This is due to the fact that the amount of water used for snowmaking on ski pistes represents a fraction of 10 to 20% of total annual precipitation, that ski pistes cover typically 10% of the surface area of catchments within which ski resorts are located, and that snowmaking equipment covers, in the case of La Plagne, 40% of the surface area of ski pistes. Therefore, in this case, snowmaking mainly leads to a moderate shift in snow cover formation and snowmelt processes, and plays, for example, a smaller role than the influence of future climate change on mountain hydrology.

This study provides an initial overview of the influence of grooming and snowmaking on river flows in a mountain catchment, which can inform future studies on water management and climate change adaptation in areas with ski tourism facilities. This study does not discuss long-term sustainability challenges of ski tourism and other aspects of the local environmental

impacts (landscape, biodiversity) of snow management, such as the construction and use of mountain water reservoirs and other earthworks in ski resorts.

## 1    Introduction

Ski resorts are a prominent component of mountain economies in many regions of the world (Europe, North America, China, Japan, New Zealand and South-East Australia) (Steiger et al., 2019; Hock et al., 2019; Vanat, 2021). Snow management, in particular grooming and snowmaking, i.e. artificial production of snow on ski pistes before and during the ski season, have become routine activities for ski resorts operations (Steiger et al., 2019). Among the various environmental concerns related to the operations of ski resorts, the hydrological impacts of snowmaking is often mentioned as an argument against their development. In particular, the use of substantial amounts of water for snowmaking is considered as an unsustainable pressure exerted on the mountain environment. It is however often argued by ski tourism supporters that water is only "borrowed" for snowmaking, and returned to mountain creeks and rivers at the time of snowmelt. Despite this contentious situation, very few scientific studies have addressed quantitatively the influence of snow management, in particular snowmaking, on the water cycle at both local and regional scale. This is however required, in order to contribute scientific information regarding the relationships between mountain water resources and ski resorts operations, at a time of intense debates regarding the transition of mountain tourism into a more sustainable pathway, taking into account climate change impacts and related adaptation options (Hock et al., 2019; Tschanz et al., 2022; François et al., 2023). The goal of the present study is to explore this knowledge gap and to shed light on the influence of grooming and snowmaking on the local hydrological cycle, in particular the river runoff downstream of watershed within which a ski resort is located.

Eisel et al. (1988, 1990) carried out pioneering studies on water losses due to snowmaking based on case studies in Colorado (United States of America), motivated primarily by water right issues. They separated water losses in two categories: "initial loss", corresponding to "consumptive water loss that occurs during the actual snowmaking process due to evaporation and sublimation" and "watershed loss", corresponding to "consumptive water loss that occurs from the time the man-made snow or ice particles have fallen on the snowpack through spring melt. This loss is due to sublimation and evapotranspiration." Eisel et al. (1988) focused on the "initial loss", which was later complemented by more recent studies carried out in Europe (Spandre et al., 2017; Grünewald and Wolfsperger, 2019). The amount of "initial losses" are estimated to be on the order of 10% of the water used for snowmaking, due to evaporation and atmospheric sublimation processes. Spandre et al. (2017) highlighted that some of the snow produced for snowmaking is not recovered on ski pistes (it simply falls besides the ski pistes), and is thus wasted from perspective of ski resort operations although it is not lost in terms of the total amount of water in the catchment. They estimated that this concerns approximately 30% of the water used for snowmaking, although it varies substantially depending on local context and meteorological conditions. Eisel et al. (1990) investigated the watershed losses for several ski resorts, by means of a comparison between observed runoff, water consumption for snowmaking and meteorological conditions, and simulated runoff. They concluded that, for the situations analysed, the watershed loss ranged between 7 to 33 % of the water used for snowmaking, after the initial loss, leading to 13 to 37 % total consumptive loss. This

indicates that, in this case, abstracting water for snow production leads to substantial net water loss for the catchments due to sublimation and evaporation of the snow cover on ski pistes. However, they clearly stated that "these results should not be extrapolated directly to other specific ski area sites because actual consumptive loss at these sites is dependent on atmospheric temperature during snowmaking, temperature, and precipitation during the winter and watershed conditions at the site." We are not aware of whether these studies were corroborated by further studies, and whether they have been used operationally, in particular for water rights applications. In a more recent study, Wemple et al. (2007) reported on the fact that snowmaking corresponded to 3 to 4 % of the total annual precipitation in a ski resort in the NE USA. Even more recently, Gerbaux et al. (2020) compared water demand for snowmaking with water resources availability in several ski resorts in the Northern French Alps, but did not address the downstream effects on the river runoff induced by snow grooming and snowmaking and potential water losses. Leroy (2015) developed an integrated modelling approach for the quantitative comparison of the amount of water withdrawals for various uses (hydropower, agriculture, domestic use, snowmaking) in a pilot ski resort in the northern French Alps, but also did not address downstream influence on river flows.

The present study explores the impact of snow management (grooming and snowmaking) on the hydrological cycle and downstream water availability, in particular seasonal patterns, at the local scale. Mostly based on modelling tools, we implement this approach as a case study in the Northern French Alps (La Plagne ski resort). We provide and analyse results at various geographical scales: (i) at the point scale at 1800 m elevation, (ii) for a study domain encompassing the entire ski resort area, and (iii) for two interconnected catchments (Frasses and Bonnegarde catchments) for which simulation results and a diverse set of observations are combined and jointly analyzed. Results are provided not only for past meteorological conditions but also under future climate conditions, for the Bonnegarde catchment.

## 2  Case study: the La Plagne ski resort

The La Plagne ski resort is the largest ski resorts in France, one the of world largest ski resorts, with over 2 million skiers visits annually (Vanat, 2021), located in the Northern French Alps. It spans an elevation range from 1250 m to 3250 m above sea level, and includes 5.28 km$^2$ of ski pistes (Ebner et al., 2021). Figure 1 shows the geographical location of La Plagne, and the geographical distribution of its pistes with and without snowmaking.

The main snowmaking technology used in La Plagne are lances. Lances are a type of snowmaking equipment, using pressurized water rather than a fan-like snowgun. French ski resorts are equipped to a great extent with such equipment. At the scale of the entire ski resort, 40.2% of the surface area of ski pistes is equipped with snowmaking, which is higher than the average value for French ski resorts (Berard-Chenu et al., 2022b).

The wider ski resort area is defined by its gravitational envelope. This concept was introduced by François et al. (2014) and later refined and implemented by further studies (François et al., 2016; Spandre et al., 2019; Berard-Chenu et al., 2022a; François et al., 2023). Indeed, there is no standard delineation of the ski resort geographical domain. The gravitational envelope corresponds to the geographical domain accessible downhill from the top of all the ski lifts and reaching the bottom of one of the ski lifts. Not all of the gravitational envelope consists of ski pistes (they cover, on average 10% of the surface area

according to Spandre et al. (2019)) and is even accessible to skiing, but it provides a representative and usable boundary for the area characterizing the ski resort, based only on the catalogue of ski lifts and the local topography. In La Plagne, the surface area of ski pistes corresponds to 9.7% of the surface area of the gravitational envelope, so that, ultimately, the gravitational envelope is covered at 3.9% by ski pistes equipped with snowmaking. Figure 1 shows the La Plagne gravitational envelope used for this study.

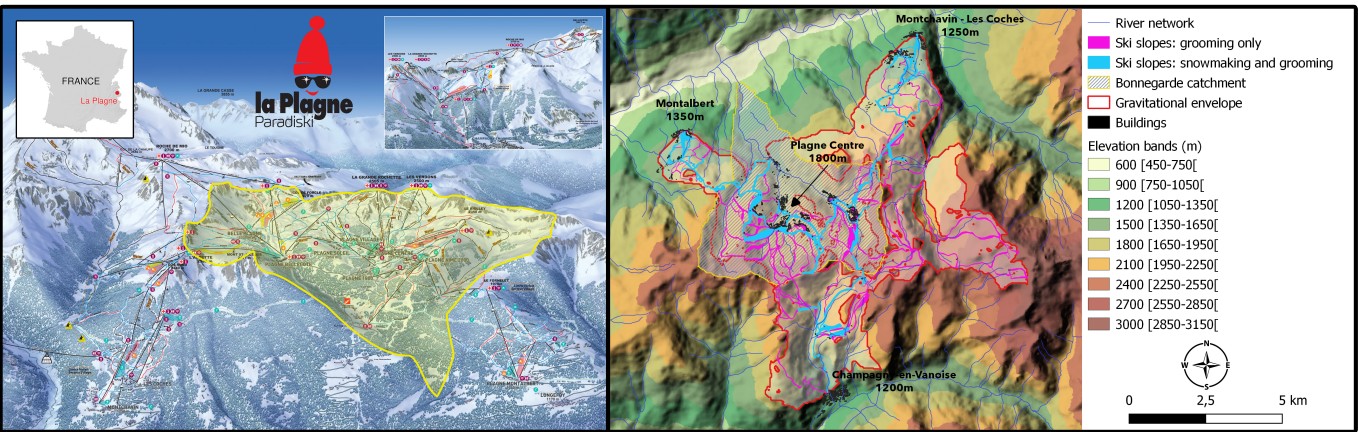

**Figure 1.** Maps of the La Plagne ski resort, with the Bonnegarde catchment in yellow. On the left hand side : location of the La Plagne ski resorts on a national map for France, landscape image provided by the La Plagne ski resort, overlay of the Bonnegarde catchment by the authors. On the right hand side, detailed map of the ski resort, showing its gravitational envelope (red solid line), ski pistes with (blue) and without (pink) snowmaking, location of main villages. Right hand-side : Hillshaded background derived from EU-DEM v1.1 (Copernicus Land Monitoring Service), further geographical information from BD Alti provided by IGN at the 25m resolution, geographical information on location and status of ski pistes provided by the La Plagne ski resort (generation of the map by the authors, see text for methodological information).

Water used for snow production in La Plagne is stored in five reservoirs (Prajourdan, Lovatière, Forcle, Pierres Blanches and Montchavin, see Figure 2) located in or near the ski resort. Water is taken from various sources (surface or underground) to fill these reservoirs. Some reservoirs are managed under a two-fold purpose, i.e. meeting water demand for snow production and water supply for domestic uses.

Figure 2 shows the sub-catchments including the ski pistes for the Bonnegarde catchment and one of its subcatchments (the Frasses catchment), which are analyzed in more details in this study. These catchments are located at the heart of the La Plagne ski resort, including main villages and ski tourism hosting infrastructures. Figure 2 illustrates the complexity of the interaction between the spatial organization of the ski resort and the local hydrological context. The Frasses catchment has a surface area of 4.71 $km^2$, and 15.2% of its surface area is covered by ski pistes. The Bonnegarde catchment, which includes the Frasses catchment, has a surface area of 23.61 $km^2$, and 19.4% of its surface area is covered by ski pistes, 36.2% of which is equipped

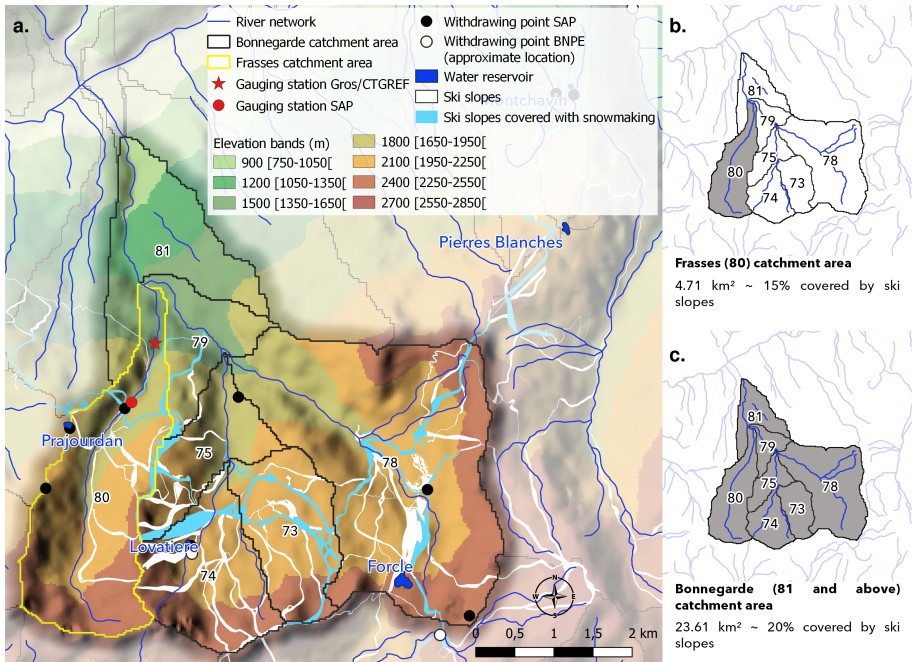

**Figure 2.** Detailed map of the Bonnegarde catchment, illustrating its various subcatchments, including the Frasses catchment. Hillshaded background derived from EU-DEM v1.1 (Copernicus Land Monitoring Service), further geographical information from BD Alti provided by IGN at the 25 m resolution, geographical information on location and status of ski pistes provided by the La Plagne ski resort (generation of the map by the authors, see text for methodological information).

with snowmaking equipment. Overall, 7.0% of the Bonnegarde catchment is covered by snowmaking equipment. The outlet of
the Bonnegarde catchment almost directly reaches the Isère river flowing though the Tarentaise valley.

## 3    Materials and methods

### 3.1    Snow cover modelling

In this study, the simulation of the natural and managed snowpack is performed by the Crocus snow cover model, which represents natural snow processes, but also grooming and snowmaking (Spandre et al., 2016b, 2019). The model accounts for all
the processes governing the internal evolution of the snowpack and its interfaces, including the surface mass balance: absorption and reflection of solar radiation, as a function of the surface albedo of the snow calculated by the model, absorption of atmospheric infrared radiation, emission of infrared thermal energy as a function of the surface temperature of the snow, latent and sensible heat fluxes at the surface of the snowpack, as a function of the surface temperature of the snow and atmospheric conditions (wind speed, temperature, relative humidity, etc.) (Vionnet et al., 2012). The implementation of snowmaking in

the model (described in detail in Spandre et al., 2016b) is consistent with typical practices in ski resorts operations (Spandre et al., 2016a), although local deviations (timing of snowmaking operations, configuration of the snowmaking units, etc.) can be observed (Abegg et al., 2020). We here use typical values rather than refining in detail the model configuration, so that the modelling system can more easily be transferred to other contexts. In the model, snow production is only possible for wind speed values below $4.2 \, \mathrm{m \, s^{-1}}$ and for wet bulb temperature values (computed using temperature and relative humidity values, as described in Spandre et al. (2016a)) below -6°C for the lances snowguns. The density of the snow produced is $600 \, \mathrm{kg \, m^{-3}}$. The representation of the grooming process includes the compaction and the mixing of snow layers (effect of the tiller), so that the snowpack on ski pistes results from the gradual mixing of snow from natural snowfall and machine-made snow (Spandre et al., 2016a). Note that the modelling system does not account for the effect of skiers on the snowpack (compaction, snow displacement) and this is implicitly represented through the effect of grooming (Spandre et al., 2016a). In early season, before the start of the main winter holiday period, production can take place between 1/11 and 15/12, only if the wet bulb and wind speed conditions are met, until a maximum of $150 \, \mathrm{kg \, m^{-2}}$ of water is converted into artificial snow, i.e. 25 cm of artificial snow at $600 \, \mathrm{kg \, m^{-3}}$, taking into account 40% water loss (combining evaporative losses (10%) and the fact that not all of the machine made snow falls on the ski pistes (30%)). Between 15/12 and 31/03 the production can take place (again, if the wet bulb temperature and wind speed conditions allow) as soon as the snow depth falls below 60 cm. There is no more production after 31/3. The model calculates the water demand corresponding to snow production, without imposing limits to the availability of water used for snowmaking. For further details, see Spandre et al. (2016a, 2019).

For areas besides the ski pistes, only natural snow cover processes are considered. Note that we do not take into account snow/forest interactions in this study, because we focus on the differences between simulations with and without grooming and snowmaking on ski pistes. Also, note that the modelling framework used in this study does not take into account wind-induced snow transport. In the modelling system used, the snowpack interacts with the underlying soil, represented by the SURFEX/ISBA land surface model (Masson et al., 2013). Thermally, the underlying soil is coupled to the snowpack through a diffusion scheme. This makes it possible to represent the insulating effect of the snow, according to its physical properties, and the thermal interaction between the snow, the ground and the atmosphere. From a hydrological point of view, several water fluxes are calculated by the model (Figure 3 ), and available at daily time step in the simulation results produced for this study. This study focuses on quantifying the impact of snow management on water availability in the watershed. The main water flux analysed and used is therefore the amount of water reaching the upper soil interface, which combines the water flux flowing at the base of the snowpack (due to snow melt) with the liquid precipitation (rain) on snow-free ground. This variable is referred to as the "total liquid water reaching soil" in Figure 3 .

### 3.2 Model simulations

In this work, consistent with many previous studies using snow cover modelling in French mountainous regions, the Crocus model is driven by meteorological conditions estimated by the SAFRAN reanalysis (temperature, precipitation, wind speed, radiation) (Vernay et al., 2022). Consistent with the geometry of the SAFRAN modelling system leading to the SAFRAN meteorological reanalysis, the simulations - named hereafter SAFRAN-Crocus simulations - are performed for 300 m altitude

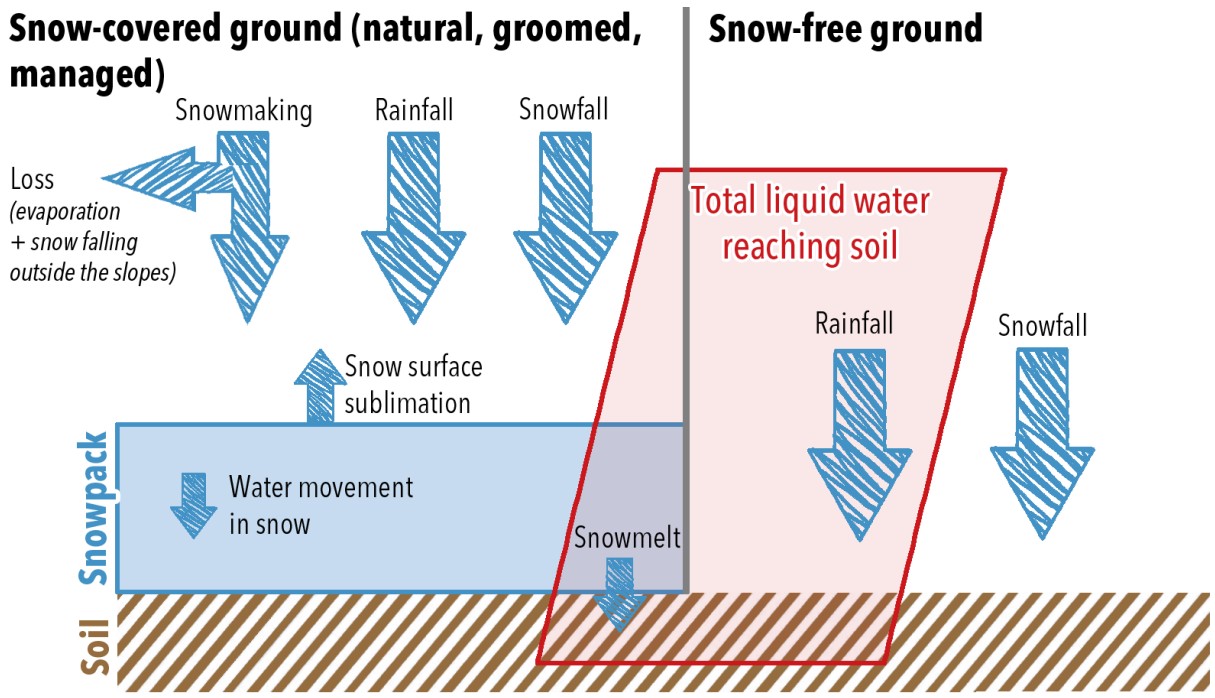

**Figure 3.** Schematic representation illustrating the total liquid water reaching soil, used in this study, calculated as the sum of rainfall over snow-free areas and liquid water fluxes exiting the snowpack.

bands within spatially homogeneous areas (massifs), on flat terrain and for slopes of 10, 20, 30, 40°, for 8 main aspects (N, NE, E, SE, S, SW, W, NW). We use SAFRAN data for the Vanoise massif, within which the La Plagne ski resort is located. The full duration of the SAFRAN-Crocus simulation used for this study spans from 1961 to 2020.

The use of SAFRAN reanalysis data feeding the Crocus model including snow grooming and snowmaking has been used as such for other studies (Spandre et al., 2019; Gerbaux et al., 2020; Morin et al., 2021; Ebner et al., 2021), which demonstrated its ability to reproduce observed snow conditions on ski pistes. In particular, Ebner et al. (2021) analyzed the performance of SAFRAN/Crocus simulations in terms of snow conditions on ski pistes at the La Plagne ski resort.

### 3.3 Geographical representation of the ski resort and catchments

Our study uses the concept of Ski resorts Representative Units (SRUs), introduced in Hanzer et al. (2020). This approach identifies unit elements characterizing the ski pistes (or their surrounding environment), taking into account their elevation and aspect and the presence or absence of snowmaking equipment. Here we use the topographical clustering of the SAFRAN reanalysis, described above. A given SRU is either concerned with natural snow (outside ski pistes), or, on ski pistes, grooming only, or grooming+snowmaking. Snow cover simulations are carried out for each SRU, and their results can be combined

according to their surface area (Hanzer et al., 2020; Ebner et al., 2021) in order to provide integrated information such as the total amount of liquid water reaching soil for a given area (gravitational envelope, catchment etc.).

Hydrological catchments located within or close to the ski resorts were derived from the high resolution digital elevation model (DEM) BD-Alti from the French National Geographic Institute (IGN), using HydroDem, a set of computer routines for processing digital elevation models from a hydrological perspective.

HydroDem was initially designed to support the scientific development of geostatistically based runoff mapping routines (e.g. Gottschalk et al., 2006). Over the years, HydroDem has been used as an aid for delineating drainage patterns in a variety of medium- and large-scale distributed hydrological model applications (e.g. Thierion et al., 2012, as in this study). HydroDem's computer code implements routines quite similar to many other softwares. The drainage pattern is built as follows: local flow directions are first estimated from the direction of local maximum slope, using 3 x 3 kernels for a smooth estimation of partial derivatives; such procedures are known to generate a suitable drainage pattern from the slopes of the Digital Elevation Model (DEM), but a number of local conflicts in the valleys, were watercourses are naturally not resolved by the raster grid cell of the DEM; a heuristic algorithm is then used to remedy them, where the lowest elevation path and the most likely way out are identified in each case. This heuristic algorithm is applied gradually, first passes accepting only modest levels of both elevation increase and connection of pool areas. Another notable trick is that the accepted elevation increase is scaled by the standard deviation of local elevation, allowing the algorithm to self-adapt in the case of sub-areas of contrasting roughness.

### 3.4 Data used for the hydrological analysis of the Bonnegarde and Frasses catchments

In addition to the snow cover modelling approach described above, we perform an in-depth analysis of the components of the hydrological cycle of the Frasses and Bonnegarde catchments (see Figure 2) based on an hydrological modelling approach and various observational data. In this area, information on hydrology is rather sparse:

1. river flow measurements have been collected between June and mid-November at a single location (corresponding to a catchment surface area of 3.85 km$^2$) by the Société d'Aménagement de la Plagne (SAP) along the Frasses River since 2013 ("Gauging station SAP" in Figure 2; 4 km$^2$);

2. a gauging station now closed ("Gauging station Gros/CTGREF" in Figure 2; 4 km$^2$) recorded discharges over two discontinuous periods (1948-1968 and in 1980). Only statistics derived from time series of daily discharge are currently available for this gauging station (average daily flows over the period 1948-1968 for Gros data and monthly flows for 1980 for CTGREF data). These discharge data were only corrected for upstream water diversion, i.e. 45 l/s each day from June to September (transfer out of the gauged catchment mainly in summer for irrigation purposes through an open channel).

In addition, we have considered time series of natural daily discharge simulated by the distributed hydrological model J2000 (Morel et al., 2023) over the period 1980-2016. The version of J2000 used here considers natural hydrological processes and was calibrated at the whole Rhone basin scale to simulate daily discharges at the outlet of medium-scale basins. The SAFRAN-France meteorological reanalysis (Vidal et al., 2010) was used as meteorological forcings for J2000. Note that, in contrast to the

SAFRAN reanalysis used for running the Crocus snow cover model, which operates for mountain massifs and 300 m elevation bands, SAFRAN-France operates on a 8 km×8 km grid.

Water abstraction data were provided by the French national database BNPE (Banque Nationale des Prélèvements quanti-tatifs en Eau, https://bnpe.eaufrance.fr/), including time series of annual withdrawals, location, use such as industry, energy production, and irrigation, and source (surface- or groundwater) since 2008. Local databases provide additional information on water management such as the location of the main withdrawing point for snowmaking ("Withdrawing point SAP"), weekly water volumes stored in each reservoir for snow production between 2007 and 2020, contribution of each reservoir to snow production over each ski piste, synoptic diagram of the water supply networks, monthly volume abstracted for and released from reservoirs for domestic water supply between 2016 and 2020. For reasons of national security, the exact location of the pumping structures for domestic water use is not publicly available and only an approximate location is indicated on Figure 2 (see "Withdrawing point BNPE").

### 3.5   Climate change simulations

In addition to simulations for past meteorological conditions based on the SAFRAN reanalysis from 1961-2020 (Vernay et al., 2022), we performed simulations under future climate change conditions. These simulations are based on one model run of the GCM/RCM pair CNRM-CM5/ALADINv6, part of the EURO-CORDEX ensemble (Jacob et al., 2014; Kotlarski et al., 2022) for the high-emission scenario RCP8.5 for a 15 years time period in the middle (2043-2057) and the end (2083-2097) of the $21^{st}$ century. These simulations are carried out using the Crocus snow cover model configuration introduced earlier, driven by the RCM model ouput adjusted with the quantile mapping adjustment method ADAMONT (Verfaillie et al., 2017) using the SAFRAN reanalysis as a reference atmospheric dataset (see Verfaillie et al., 2018, for more details). These simulations are processed similarly to the SAFRAN-driven simulations, and are meant to illustrate how the results obtained in past and current climate conditions are potentially modified under a warmer climate - although this study does not address the diversity of the sources of uncertainty involved in climate change simulations. Note that the multi-annual temperature and precipitation values at the scale of the French Alps at 1800 m elevation, for the ADAMONT-adjusted results of the CNRM-CM5/ALADINv6 GCM/RCM pair, correspond the median of an ensemble of 19 adjusted GCM/RCM pairs, for the time period 2090-2099 under RCP8.5 (Monteiro et al., 2022). This choice of GCM/RCM pair can therefore be considered as representative of the mean climate change signal, pending further investigations using an ensemble of climate model simulations.

## 4   Results

### 4.1   Hydrological effects at 1800 m elevation

Grooming and snowmaking influence the characteristics of the snowpack and this translates into its hydrological behaviour. As an example, Figure 4 displays, for the winter season 2019-2020, the time evolution of the snow depth for natural, grooming and grooming+snowmaking simulations at 1800 m elevation in the Vanoise massif, typical of the situation of the La Plagne ski

resort. Several key features are highlighted on this figure. First of all, it shows that simulations with groomed snow result in lower snow depth, due to the compaction induced by the grooming process.

At the end of the season, the snow melt-out date occurs later in groomed snow simulations compared to natural snow simulations. This effect was described and analyzed in Spandre et al. (2016b). It is due to the generally higher snow density due to grooming, thus higher thermal conduction through a groomed snowpack, leading to more efficient loss of energy from the snowpack (and underlying ground), especially at night, progressively inducing colder conditions in the snowpack (and underlying ground) and delaying melt processes.

Simulations with snowmaking lead to higher snow depth values since the beginning of the season (due to additional snow mass added to the snowpack through snowmaking). The melt-out date is further delayed compared to grooming-only simulations (but comparatively less than between groomed and natural snow simulations), due to the additional mass that needs to be melt until complete melt-out. The cumulative total precipitation over the entire hydrological year matches exactly the cumulative total liquid water reaching soil for the natural snow and grooming snow simulations, which indicates that evaporation from the snowpack plays a negligible role in the simulation results. In contrast, when snowmaking is taken into account, the cumulative value of total liquid water reaching soil exceeds total annual precipitation, due to the addition of snow through snowmaking.

## 4.2  Hydrological effects over the gravitational envelope of the ski resort

The aggregated influence of grooming and snowmaking on liquid water reaching soil, at the scale of the entire La Plagne ski resort gravitational envelope (see Figure 1) is examined in this section. Note that this does not bear direct hydrological relevance because the ski resort is located on independent catchments on different aspects of the mountain over which the ski resort develops. Here, we also assume in this illustrative case that 100% of the ski pistes are covered with snowmaking.

Figure 5a shows the aggregated amount of liquid water reaching soil for the hydrological year 2019/2020, aggregated only over the ski pistes, i.e. neglecting all areas surrounding the pistes. The figure displays daily (instantaneous) and cumulative (from August $1^{st}$ 2019 to August $1^{st}$ 2020) values. In this case, the simulation shows that for most of the winter season, the simulations including grooming (with and without snowmaking) lead to lower values of the total liquid water reaching soil, especially during the winter, than natural snow simulations. In terms of cumulative amounts, not only the cumulative values for the natural snow cover simulation increases steadily during the winter season due to higher basal snowmelt than for groomed or grooming+snowmaking snow simulations, but the late season melt peak starts earlier.

Natural and grooming snow cover simulations converge, in terms of cumulative total liquid water reaching soil, in early June, and the cumulative values remain very close until the end of the season. Simulations including snowmaking converge around the same date, but then keep increasing over the end of the season, exceeding the total amount of liquid water reaching soil by about $800,000\,\mathrm{m}^3$, corresponding to 11.1% of the annual total liquid water reaching soil. Aggregated over the ski pistes only, the results reveal that grooming (with or without snowmaking) exerts a substantial influence on the water balance. This is mostly due to the fact that grooming leads to essentially suppressing winter snowmelt (leading to deficits, reaching more than

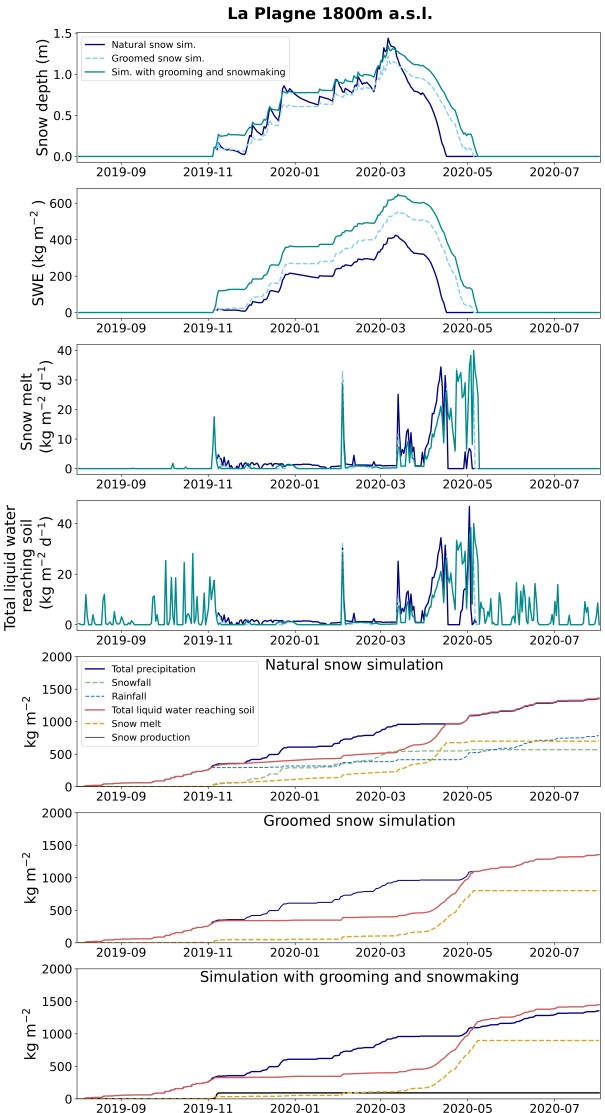

**Figure 4.** Model simulations at 1800 m elevation, representative for the La Plagne ski resort, with natural snow simulation (blue solid line), groomed snow simulation (dashed light blue line) and the simulation with grooming and snowmaking (green solid line), for the hydrological year 2019-2020. From top to bottom, the panels display snow depth, snow water equivalent (SWE), snow melt, liquid water reaching soil (snow melt flux + rainfall on snow-free ground), and cumulative water fluxes for the three simulations. These show total precipitation (solid blue line), snowfall (green dashed line), rainfall (blue dashed line), liquid water reaching soil (red solid line), snow melt (orange dashed line). The last panel further shows water demand for snowmaking (black solid line).

-40%, on cumulative total liquid water reaching soil in April), while snowmaking influences mostly the water balance towards
the end of the melt season.

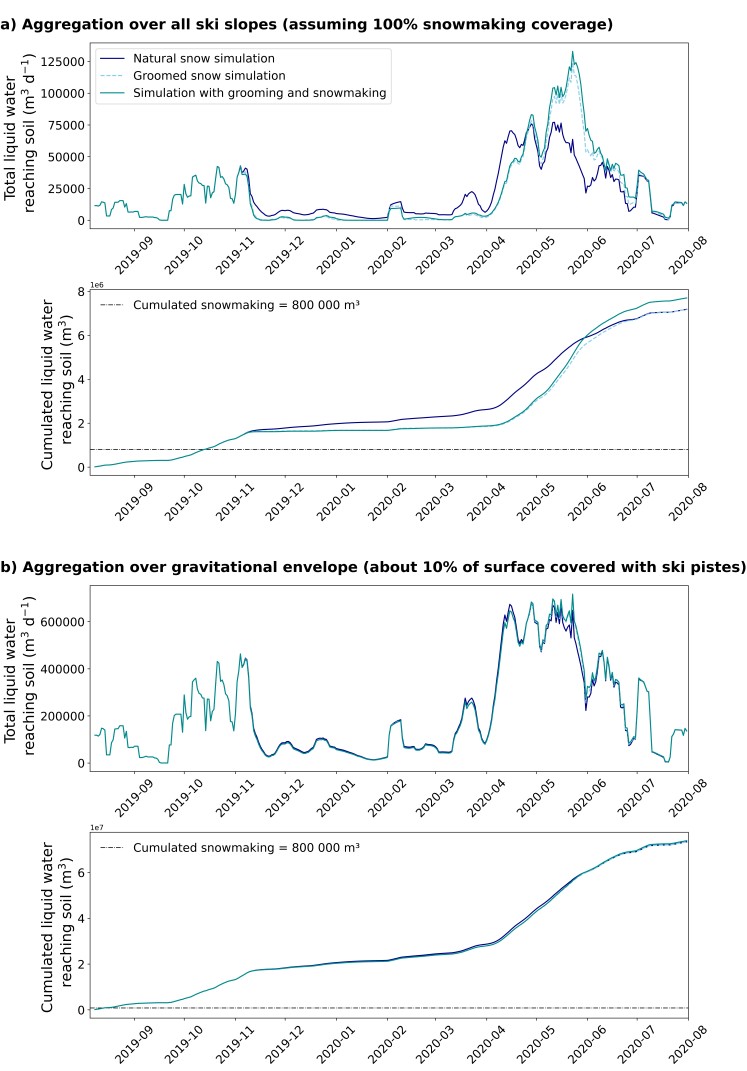

**Figure 5.** Aggregation over all ski pistes (a) and for the entire ski resort gravitational envelope (b) of the daily liquid water reaching the soil and cumulative liquid water reaching soil, respectively, for the natural snow (blue solid line), grooming snow on ski pistes (green dashed line) and grooming and snowmaking on ski pistes (green solid line).

Figure 5b provides similar types of results as Figure 5a, but the aggregation now extends to the entire gravitational envelope of the ski resort. The differences between the configurations is smaller, with a deficit of cumulative total liquid water reaching soil on the order of -5% as of April for the grooming and grooming+snowmaking configurations on ski pistes, compared to a simulation when natural snow cover processes only are considered for the whole domain. In the case when snowmaking is taken into account (assuming full coverage of the ski pistes with snowmaking equipment), the excess in total annual liquid water reaching soil amounts 1.1% of the value for natural snow processes only. The difference between the results of the two

scales of aggregation (5a and 5b) is directly linked to the surface area covered with ski pistes, which is generally on the order of 10% in French ski resorts (François et al., 2016), and specifically 9.7% for the La Plagne ski resort.

## 4.3 Hydrological effects at the catchment scale

The goal of this section is to assess the water footprint of a specific use (snow management in ski resorts), using natural resources as a reference. Four sources of available river discharge data are used to characterise natural resources, as introduced in section 3.4, focusing on the Frasses and Bonnegarde catchments. Figure 6 shows the rivers discharge of the Frasses River at the Gros/CTGREF gauging station (see Figure 2) from various observational and modelling sources. Note that this gauging station corresponds almost to the outlet of the Frasses catchment.

The SAP data are both incomplete (e.g. some hydraulic conditions did not allow current meter measurements, see filled triangle point-up in Figure 6) and influenced by upstream abstractions (e.g. the Prajourdan reservoir has been supplied by the water intake on the Frasses River since 2001). The SAP data were thus not considered as a reference.

As shown on Figure 6, the other three data sources suggest a nival (snow-dominated) river flow regime with a peak in June due to snowmelt. They show similar hydrographs and time series are highly correlated (for example, $R^2 = 0.83$ between 280 Gros and CTGREF data on a monthly scale, and $R^2 = 0.73$ between Gros and J2000 data on a daily scale). However, they are not perfectly consistent in terms of annual volume. The annual flow is equal to 49 $L\,s^{-1}$ according to the J2000 model, whereas measurements at the Gros/CTGREF gauging station estimate an annual flow of around 200 $L\,s^{-1}$. The outputs of the J2000 model do not appear to be usable because of the bias identified in the annual flow, perhaps due to the excessively coarse resolution of the meteorological forcings (64 $km^2$) in relation to the size of the catchment. In contrast, the aggregation 285 of the "total water reaching soil" from the SAFRAN-Crocus for the Frasses catchment (using simulations of natural snow around ski pistes and only taking grooming into account on ski pistes, and spanning the time period 1961-2020), shows a better agreement with the CTGREF and Gros datasets given that the annual flow simulated by SAFRAN-Crocus is equal to 162 $L\,s^{-1}$. However, Crocus-based results show too low discharge values (even unrealistic zero-flow conditions in some cases, see the red envelop in Figure 6) in wintertime leading to $R^2 = 0.64$ between SAFRAN-Crocus and CTGREF, which we interpret as 290 the consequence of the lack of a genuine hydrological modelling framework in this case, leading to ignoring e.g. groundwater processes sustaining winter baseflow. We also note that the variables used from the SAFRAN-Crocus simulations do not take into account evapotranspiration. However, this figure shows that the catchment-level aggregation SAFRAN-Crocus simulations of total water reaching soil bears hydrological relevance compared to observed datasets, and can be used further, especially for comparing simulations performed using different configurations in terms of snow-related processes.

Finally, mean daily flows from Gros data are used later as reference values for natural resources rather than CTGREF data, which are only representative of a single year. It should be noted that the observation period linked to the Gros records is old (1948-1968) but nevertheless seems still representative of the recent period (based on a comparison, even though it is questionable, with the mean annual hydrograph provided by J2000 over the period 1980-2016 and with the measurements made by SAP in autumn when abstractions are lower).

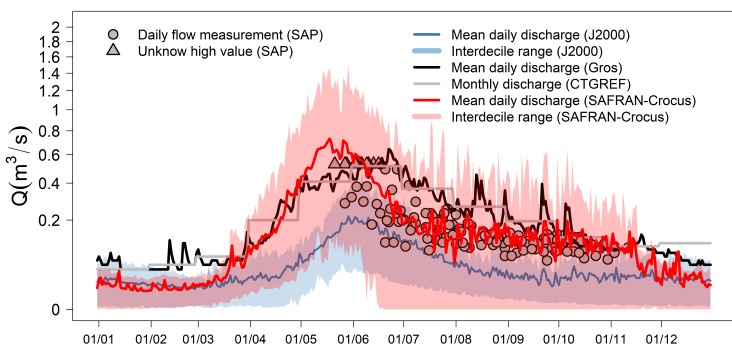

**Figure 6.** Annual hydrographs derived from available discharge observation data and simulation results for the Frasses catchment. Data from the J2000 model are those simulated at the gauging station Gros/CTGREF: the shaded area indicates the interdecile range, and the mean hydrograph is in dark blue. Monthly discharge for 1980 from CTGREF are displayed in grey. The Crocus results correspond to the aggregation of the total water reaching soil for the Frasses catchment, spanning the time period from 1961 to 2020. The grey points are flow measurements made by SAP, identified on the x-axis by the Julian day of the recordings. The y-axis displays a square-root scale.

We now introduce results focusing on the two catchments Frasses and Bonnegarde (Frasses is a sub-catchment of Bonnegarde, see Figure 2).

The approach considered here is purely empirical: it is based on a water balance model applied to observed data and results from snow cover simulations at different time scales (from week to year).

Only first-order estimates of the water footprints developed along the river system can be expected due to limited direct information on the snowmaking process, water abstraction and consumption. For example, the water balance is partly biased by non-concomitant data periods (dating back to the 1940s and 1960s for water resources, and after 2007 for water abstraction), whereas the demand for water for snowmaking and reservoir management is directly linked to the meteorological and hydrological conditions of the year.

The Bonnegarde catchment is ungauged. There is only one gauged catchment in the neighbourhood (the Frasses catchment), and the hydrological distributed model J2000 fails to reproduce reliable discharges at the gauging station Gros/CTGREF. The drainage-area ratio method was thus applied considering that the runoff per unit drainage area is equal over the ski resort catchment.

The other working assumptions for the calculation of the water uses are the following:

– In the following, values are negative when they relate to abstraction, loss or consumption of water.

– Monthly volumes for water domestic use are evenly distributed within the month at weekly time step. When water used for domestic purposes is abstracted and consumed in the catchment, the water returns to the river system with an efficiency rate of 85% (https://www.services.eaufrance.fr/commune/73150). When the water is returned outside the catchment area, a total loss of 100% is assumed.

– Only the water for snow production (WR) is taken into account in the calculation of the water balance. The impact of the Prajourdan reservoir has been included in the calculations for both catchments. The other two reservoirs Forcle and Lovatière have an influence limited to the Bonnegarde catchment. Transfers between reservoirs do exist but are neglected.

– Values for the volumes of snow produced (SP) are given by SAP for each equipped slopes and are aggregated at the catchment scale in proportion to the length of the equipped slope crossing the catchment.

– The effect (SGE) of the snow grooming and snowmaking is provided by the simulated values of total liquid water reaching soil described above. This effect is given by the average difference between natural snow and grooming+snowmaking simulations of total liquid water reaching the soil at the weekly time scale over the period 1961-2020 and for the entire ski resort gravitational envelope. The obtained values are then divided by the total volume to provide a dimensionless pattern for each ski piste. The local impact of grooming and snowmaking is calculated for each ski piste as the sum of this pattern multiplied by 60% of the total volume of water used for snowmaking, and the mean pattern of total liquid water reaching soil simulated considering natural snow, multiplied by 30% of the total volume of water used for snowmaking. A total of 10% is considered to be definitely lost by evaporation and sublimation of the water droplets during the snowmaking process.

– The hydrological alteration due to grooming and snowmaking including reservoir management is the sum of WR and SGE.

The results are presented starting the series at week 31 (which includes August $1^{st}$) on Figures 7 and 8. The colours on these two figures were chosen to better distinguish the different types of water use (green and blue for domestic use and for grooming and snowmaking, respectively) from natural surface water availability (black). The shades of blue distinguish between the solid (S, dark blue) and liquid (L, light blue) phases of water. Inter-annual averaged values for week 53 are added to those for week 52. Because of the skewed distribution of mean weekly discharges, the y-axis is represented with a square root scale. Negative values represent water losses. The water balance is computed at the seasonal and yearly time scales (Table 1), from data with distinct periods of availability (e.g. 2007-2020 for reservoir management, 2014-2020 for observations of water used to produce snow, 2016-2020 for observations of domestic water use, and 1948-1968 for observations of water resources (river discharge), 1961-2020 for snow cover modelling including the effect of grooming and snowmaking). Hence, the results can be viewed as a "typical" water year, compositing information from different time periods, due to the heterogeneities of the data sources. The "volumes" of snow-related quantities in 7d and 8d correspond to snow water equivalent.

Figures 7a and 8a show that the seasonal pattern of river flows for both catchments is typical of snowmelt-dominated regimes with high flows in spring or early summer and winter minima caused by freezing. The different types of water use alter in different ways the river flow regime. The influence of water supply (Figures 7b and 8b) is highly variable in both space and time even within a rather small ski area. According to the schematic diagrams of water supply, all of the water amounts abstracted for domestic uses within the Frasses catchment are consumed and returned to rivers outside the catchment. One part of water supply is abstracted and consumed within the Bonnegarde catchment, and the other part is abstracted within

and exported directly from the sources to downstream areas. Water abstraction for domestic use is closely linked to tourist activities: two peaks occur, one during the ski season (December to April) and the other during the summer (July and August) (Figures 7b and 8b).

The patterns of weekly water abstraction for snowmaking (Figures 7c and 8c) is quite similar for the two catchments. However, analysing abstracted volume at the catchment scale hides heterogeneities. The seasonal patterns differ from one reservoir to another, and from one year to another, due in particular to technical constraints (availability of an overflow on intakes shared with the water supply) and regulatory constraints (compliance with ecological flows). Withdrawals to the Prajourdan reservoir (see Figure 1) are concentrated in autumn and winter, while those to the Lovatière reservoir are evenly distributed throughout
the year and those to the Forcle reservoir are concentrated in summer.

       This analysis enables to study the influence of grooming and snowmaking on river flow. The results show that they modify river flows throughout the year except in summer. Snow is first produced to provide a ground layer of snow at the beginning of the winter season (November-December), and afterwards (up to end-March) to maintain a minimum snow depth for skiing (Figures 7d and 8d). Melting processes as described in the previous section are delayed and the snowmelt peak is observed
between April and May (Figures 7e and 8e).

       The effect of the grooming and snowmaking operations on river flow regime (Figures 7f and 8f) is the sum of individual alterations. Alterations are mainly due to abstractions for filling reservoirs. Grooming and snowmaking modify the natural flow regime of rivers crossed by the ski area in different ways. At the catchment scale, the first modification is linked to the process of filling the reservoirs for snow production; the filling up is constrained by regulations on ecological flows and by the
water availability in reservoirs shared with other uses (water supply for domestic use in the La Plagne ski resort). The filling starts in autumn, before the first phase of snow production. The second effect is related to the presence of groomed snow on the ski pistes and on the hillsides. In winter, grooming leads to more severe low flows while, later, river flows increases due to increased amount of snowmelt water in spring. When points and sources for water abstraction, and equipped ski pistes, are located within the same basin, the effects are only apparent at the seasonal scale (machine-made snow then plays the role of an
additional buffer reservoir with storage in autumn and release in next spring) and rather neutral on the annual scale in terms of water resources.

       In the two catchments, snowmaking causes water losses because water abstracted for production is transferred to other nearby areas: the water balance is overall negative for both catchments (see the last column in Table 1), but it is overall positive for the surrounding areas, which benefit from the water imported from the reservoirs (not shown here). Note also that the
pressure on the resource is not evenly distributed within the year. The pressure is highest in winter, with a contribution of snowmaking use to the total withdrawals of around 20% of the resource for the Frasses catchment and around 10% for the Bonnegarde catchment. The balance for domestic water use (first column inTable 1) is also negative for the same reasons as for snowmaking use (transfer to neighbouring areas). The pressure on water resources is similar in terms of annual volume, but less contrasted over the year. The pressure from domestic uses is at its highest during the touristic season, as it is for snowmaking.
This is the low-water period in mountainous areas and this use may compete with water supply for domestic use (Leroy, 2015).

**Table 1.** Water balance at the catchment scale ($\times$ 1000 m$^3$)

| | River flow | Domestic water use | Abstraction for reservoir filling to produce snow | Snow production | Grooming and snowmaking effect | Alteration due to grooming and snowmaking |
|---|---|---|---|---|---|---|
| | | | Frasses catchment | | | |
| DJF | 518.9 | -35.3 | -91.4 | 34.1 | -10.2 | -101.6 |
| MAM | 2210.2 | -25.7 | -5.7 | 0.7 | 13.9 | 8.1 |
| JJA | 3476.8 | -25.4 | -19.3 | 0.0 | 50.0 | 30.8 |
| SON | 1261.5 | -13.2 | -27.6 | 23 | -1.6 | -29.3 |
| Year | 7467.4 | -99.6 | -144 | 57.9 | 52.1 | -91.9 |
| | | | Bonnegarde catchment | | | |
| DJF | 2600.9 | -67.8 | -215.5 | 113.2 | -30.0 | -245.5 |
| MAM | 11079.2 | -44.9 | -26.5 | 2.9 | -11.2 | -37.6 |
| JJA | 17428.4 | -36.1 | -67.6 | 0.0 | 212.5 | 144.8 |
| SON | 6323.7 | -19.2 | -92.2 | 64.3 | -9.0 | -101.2 |
| Year | 37432.2 | -168.0 | -401.8 | 180.4 | 162.3 | -239.5 |

## 4.4 Hydrological effects under past and future climate in the Bonnegarde catchment

Figure 9 illustrates the hydrologicals effects of grooming and snowmaking of the Bonnegarde catchment under past climate (1986-2015, based on the SAFRAN reanalysis) and for two future time periods in the middle (2050: 2043-2057) and the end (2090 : 2083-2097) of the 21$^{st}$ century using the adjusted GCM/RCM pair CNRM-CM5/ALADINv6 for the high-emission scenario RCP8.5. Note that the Bonnegarde catchment (see Figure 1) has a surface area of 23.61 km$^2$, and 19.4% of its surface area is covered by ski pistes, 36.2% of which is equipped with snowmaking equipment. Overall, 7.0% of the Bonnegarde catchment is covered by snowmaking equipment.

Figure 9a shows the cumulative total water reaching soil over the Bonnegarde catchment for three different configurations (natural snow over the full catchment, grooming on ski pistes and natural snow around, and gromming on ski pistes, snowmaking on equipped ski pistes, and natural snow around), using a similar representation as on Figure 5.

Figure 9c shows the evolution of the total liquid water reaching soil over the catchment, using only natural snow conditions, showing the typical trend towards an earlier snowmelt peak under a warming climate in mountainous catchments (Hock et al., 2019). Under past climate, the peak flow occurs in May, and this shifts to April in future climate projections. Figure 9d shows similar information, but accounting for grooming on ski pistes. A similar pattern is observed. The alteration, i.e. the monthly difference compared to the natural snow conditions, is shown in absolute terms on Figure 9f and relative terms on Figure 9i. The main effect, as shown in previous sections, is a lower snowmelt flux during the wintertime, low flow period, on the order of -10 to -20%, and a higher value later in the season, consistent with previously introduced results. Note that this alteration

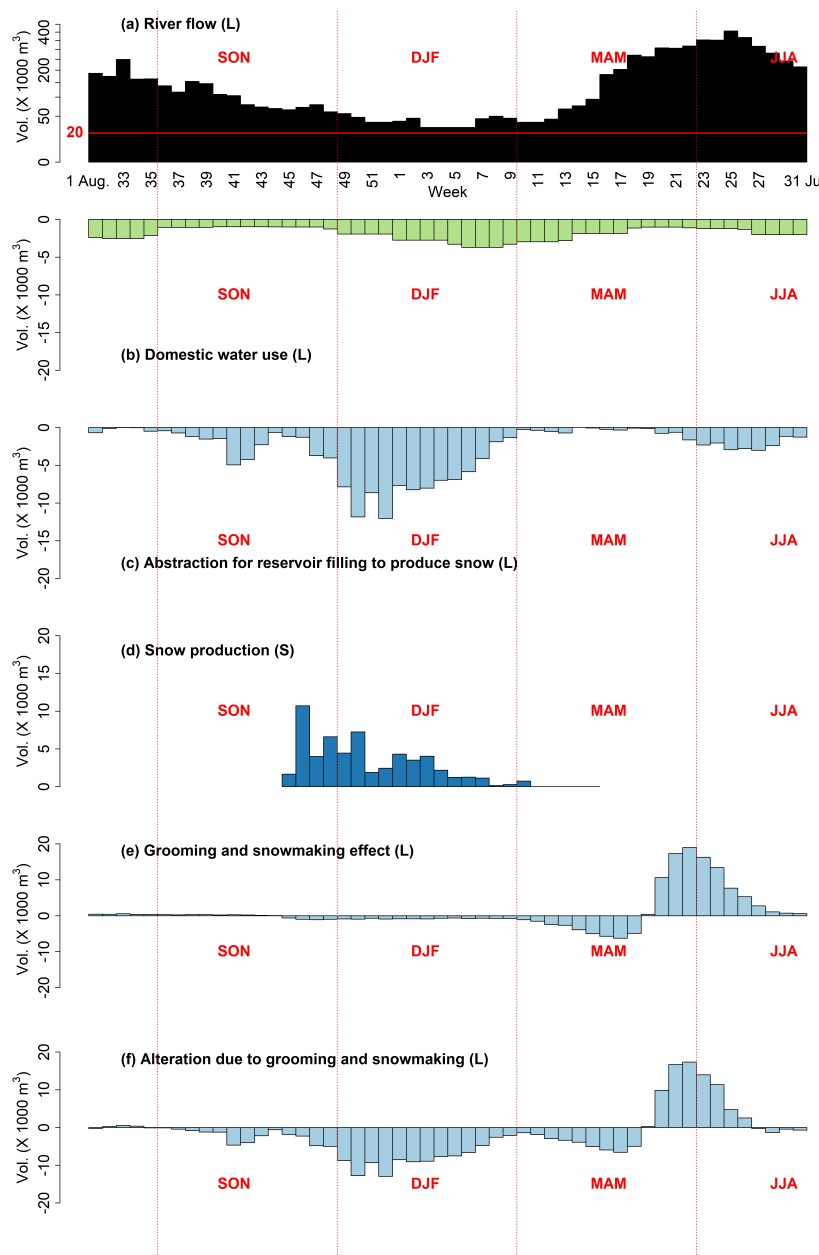

**Figure 7.** Synthesis of water resources, abstraction and consumption for the Frasses catchment including water phases (S: solid; L: liquid). The red line on panel (a) represents the ecological flow for the corresponding catchment.

corresponds almost exactly to the fraction of the catchment covered by ski pistes, and is directly due to the quasi-suppression of wintertime snowmelt on groomed ski pistes (with or without snowmaking).

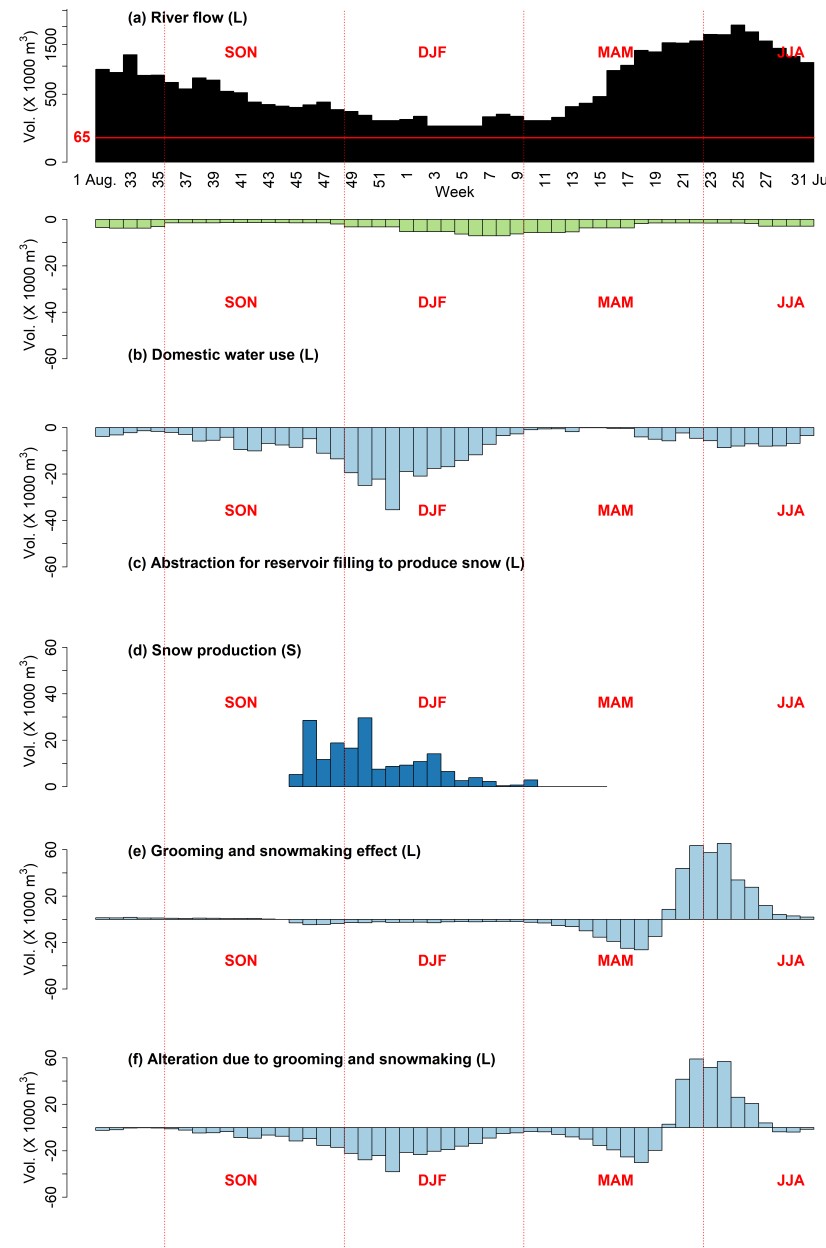

**Figure 8.** Same as Figure 7 but for the Bonnegarde catchment.

Figure 9b shows the total water demand for snowmaking on ski pistes equipped with snowmaking in this catchment. It shows that, for the early-season period (especially November), the amount of snowmaking decreases in a warmer climate, consistent with the fact that conditions favourable for snowmaking become less frequent (Spandre et al., 2019; Morin et al., 2021).

However, in a warmer climate, water demand for snowmaking increases in the second half of the season, due to increasingly frequent situations with low snow depth values (below 60 cm), triggering snowmaking in the model simulations as soon as wet bulb temperature values are below the threshold. This mostly occurs in January. Overall, in a warmer climate, total water demand tends to be larger than in the reference simulations 1986-2015, although the water demand is not larger for end of century than for mid-century, most certainly due to the fact that "cold-windows" appropriate for snowmaking become increasingly rare and inhibit the capacity to operate snowmaking facilities.

Figure 9e shows the aggregated total liquid water reaching soil, in the situation where snowmaking is implemented on equipped ski pistes of the Bonnegarde catchment. It does not show major differences with simulations ignoring snowmaking (Figure 9c).

Figure 9g shows the difference, for each future time period, between the simulations with and without snowmaking (on ski pistes equipped with snowmaking), in absolute terms, and for Figure 9j in relative terms. It shows that the impact of snowmaking on the monthly total liquid water reaching soil is maximum in June under past climate conditions, and progressively shifts towards earlier time period under a warmer climate, peaking to 10% difference, on average, in May, at the end of the $21^{st}$ century under RCP8.5.

Figures 9h and 9h show the absolute and relative alteration between the simulations accounting for grooming and snowmaking, and the fully natural snow conditions. There, the two effects of grooming and snowmaking combine. Grooming leads to lower low flows in wintertime, while snowmaking leads to increased snowmelt amount in April, May and June, depending on the climate context (earlier peak in alteration in a warmer climate).

This very limited change is mainly due to the fact that, while the annual amount of snowmaking shows a substantial increase in the simulations, it corresponds to an overall limited fraction of the local water resources at the scale of the catchment. The impact of snowmaking on monthly total liquid water reaching soil remains lower than shifts in (natural) snow melt seasonal patters due to climate change, both in terms of annual and monthly values, and does not substantially counteract the effect of climate change on the shift in snow melt pattern, even if snow management leads to a delayed snow melt on ski pistes.

## 5 Discussion

The present study explores the influence of grooming and snowmaking on the hydrological regime of alpine catchments within which ski pistes are located. It introduces a method for such an investigation, based on numerical modelling, which is illustrated and demonstrated using the case study of the La Plagne ski resort, specifically. Prior to discussing the implications of the results, we first highlight some of the key limitations of this work.

### 5.1 Limitations

The current study introduces a method, based on existing snow cover modelling tools designed to carry out numerical simulation of the snow cover in ski resorts (Spandre et al., 2016b; Hanzer et al., 2020; Ebner et al., 2021), in order to assess the influence of snow management on the hydrology of associated catchments. Using such tools enables to simulate the hy-

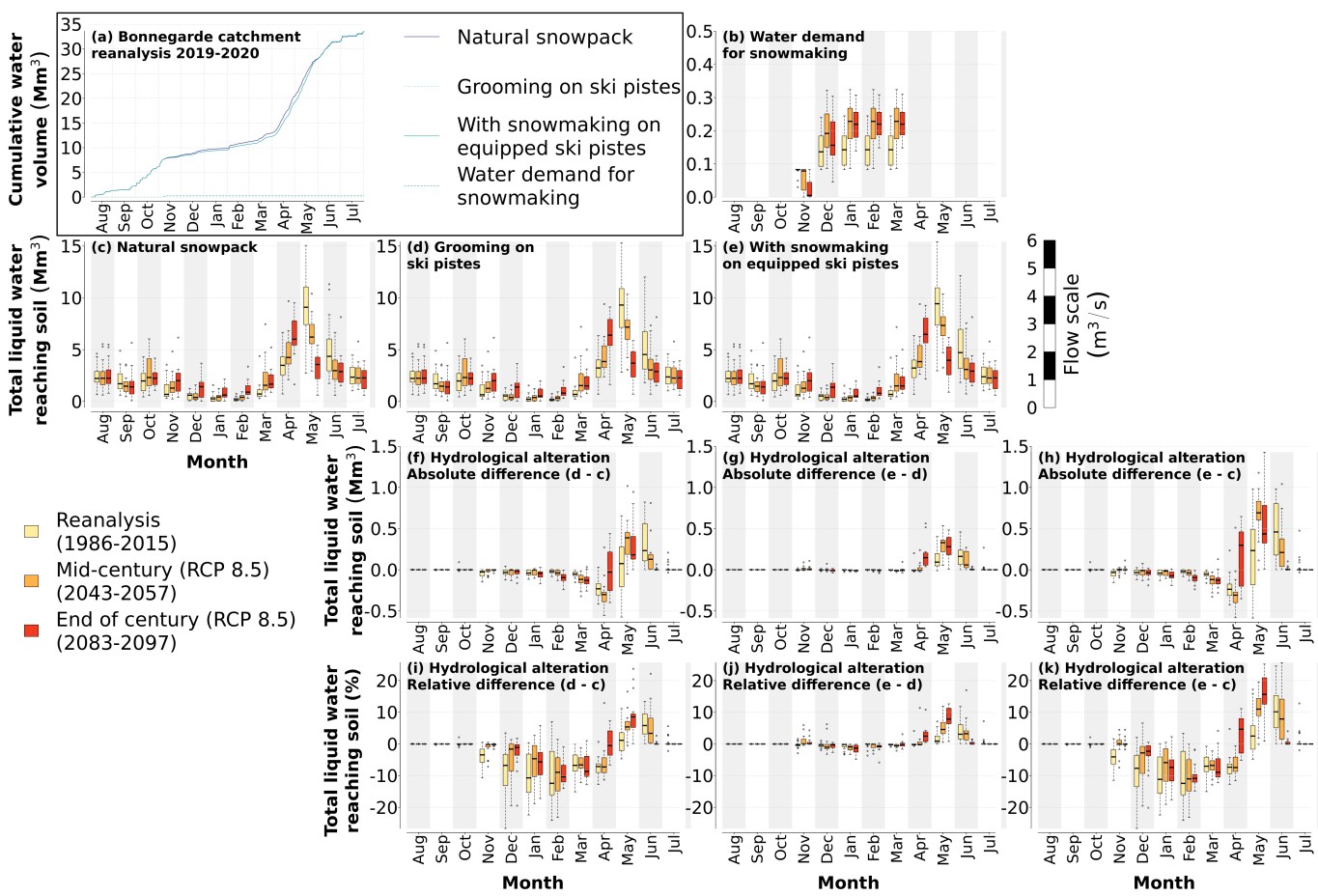

**Figure 9.** Overview of the hydrological alteration due to grooming and snowmaking in the Bonnegarde catchment, based on the SAFRAN-Crocus simulations. Simulations take into account either natural snow only (c), grooming on ski pistes and natural snow around the pistes (d), and grooming on ski pistes and snowmaking on the parts of the ski pistes equipped with snowmaking equipment, and natural snow around the pistes (e). Here we do not take into account the water abstraction used for snowmaking, similar to Figure 5. Panel (a) shows the aggregated total water reaching soil for the whole catchment for the hydrological year 2019-2020 (similar to Figure 5). Panels (b) to (k) show various components and visualizations of the local hydrological cycle under past climate conditions (reanalysis, 1986-2015) and using climate projections (RCP8.5, centered on 2050 (2043-2057) and 2090 (2083-2097)). Panels (c) to (e) show the distribution of monthly agregations of total water reaching soil under these three climate contexts. Panel (b) shows the cumulative water demand for snowmaking under these three climate contexts. Panels (f) to (h) show the hydrological alteration, here computed as the difference between monthly aggregated total water reaching soil between different configurations and with various references (natural snow only, or taking into account grooming on ski pistes). Panels (i) to (k) show the relative alteration, i.e. same as panels (f) to (h) but relative to the corresponding reference values.

drological role of the snow cover, with and without grooming and snowmaking, yet this does not pretend to encompass the complexity of hydrological processes that a detailed hydrological model would allow to account for. For example, our focus on the variable "total water reaching soil", obtained as the sum of (i) liquid water flowing at the bottom of the snowpack and (ii) rainfall on snow-free ground, and aggregated over a given catchment, does not take into account other key hydrological processes such as subsurface flow and evapotranspiration. The absence of subsurface flow is most probably responsible for the

fact that the wintertime river runoff in the Frasses river from this modelling approach, as shown on Figure 6, is underestimated compared to available observations, in particular cases of discharge values equal to zero. However, the simulations compare rather favourably in terms of discharge at other time periods of the year, which indicates that the proxy variable "aggregated total water reaching soil" is relevant to assess the differences between two model simulations using various configurations. Despite these limitations, our modelling approach also brings new results, that could not be obtained from a classical hydrological

model. For example, our study illustrates the substantial role of grooming on the physical behaviour of the snowpack and in particular basal melt, which is essentially suppressed due to grooming. Such results would not be revealed using snowpack models not accounting for snow physics, in a way sufficiently detailed so as to account for the influence of the grooming process. For example, snowpack components of hydrological models, which are formulated in terms of a single snow mass reservoir (snow water equivalent), with a melt rate formulation only dependent on air temperature and, potentially, radiation (temperature-index

models), cannot account for the different physical behaviour of snow on ski pistes, compared to the surrounding natural snow cover. Likewise, our approach enables to explicitly account for snowmaking on the snowpack properties. Overall, our approach provides a quantitative appraisal on the influence of grooming and snowmaking on the water budget of the related catchment.

The current study can be used to trigger the development of more complex, integrated modelling approaches, within which snow cover processes can be represented consistent with the approach described here, while hydrological processes can be

represented with more details. This also applies to the representation of the balance between the water demand induced by snowmaking, and the water availability from various sources, including the fact that water can be withdrawn from one catchment, stored, and used in another catchment. However, developing models capable of representing this full complexity through a genuine, two-ways coupling between snow cover simulation, including snowmaking, and the hydrological functioning of mountainous catchments at the local scale of a few hundreds of meters of horizontal resolution, represents a very challenging

endeavour (Hanzer et al., 2014). Furthermore, this approach would be difficult to transfer to other locations. In contrast, we suggest that our approach can be further tested by other teams, and used in combination with other data sources, in order to better assess the balance between water demand and water availability, and local and downstream impacts, and contribute to better informing discussions relevant to water management in mountain regions.

Hydrological model simulations derived from J2000 were assessed but not used due to strong biases. Nevertheless, future

works could make use of such models using more adequate meteorological driving data in mountainous areas.

## 5.2 Implications

Notwithstanding the limitations highlighted above, this study brings novel insights into the influence of snow management on the hydrology of mountain catchments within which ski resorts are located. While the detailed results are provided and

directly applicable to the case study of the ski resorts of La Plagne, in the Northern French Alps, this study bears implications for the related scientific domain, and for water resources management and ski resorts in mountainous areas. First of all, our results highlight that the hydrological effect of grooming and snowmaking scales, to a large extent, with (i) the surface area covered with ski pistes, driving the deficit in winter snowmelt due to grooming, and the (ii) fractional snowmaking coverage, driving the amount of excess snowmelt due to snowmaking (and the corresponding amount of prior water abstraction). Further, the influence of snowmaking is directly related to the ratio between the amount of machine made snow produced and total precipitation (aggregated over the surface equipped with snowmaking). In fact, our study supports the following equation describing the annual-scale alteration of water resources affected by snowmaking:

$$A = \frac{x_{snowmaking} \times P_{snowmaking}}{P_{annual,pistes}} \times \frac{S_{pistes}}{S_{catchment}} \times R_{snowmaking} \tag{1}$$

where $A$ is the annual-scale alteration due to snowmaking (unitless, in relative terms), $P_{snowmaking}$ is the annual amount of water used for snowmaking, $x_{snowmaking}$ is the efficiency of the liquid-water-to-snow mass conversion (90% typically), $P_{annual,pistes}$ is the annual precipitation amount falling on ski pistes, $S_{pistes}$ and $S_{catchment}$ are the surface area of the ski pistes and catchment, respectively, and $R_{snowmaking}$ represents the fraction of the ski pistes surface area equipped with snowmaking.

In the case of the La Plagne ski resort studied here, the annual snow production corresponds to 10 to 20% of the annual precipitation on ski pistes. Ski pistes cover about 10% of the surface area of the entire ski resort. Taking into account the snowmaking fractional coverage (on average 40% in La Plagne), this leads to 0.4 to 0.8% of the total water discharge aggregated over an entire year. The upper value can be increased at the local scale, in cases where the proportion of ski pistes in the catchment is higher (up to 20%) and the snowmaking fraction coverage is higher (up to 80% typically), or if the amount of water used for snowmaking corresponds to a higher fraction of annual precipitation. In the case where the water is taken from a different catchment, this leads to a net increase in water discharge in the catchment where the ski resort is located, compared to an undisturbed case. In cases where at least part of the water abstraction takes place in the same catchment, then the net alteration at the annual scale is smaller than $A$, due to the compensation, to a large extent (except the 10% evaporation, see below), between water uptaken for snowmaking and water resulting from the melt of the produced snow on ski pistes.

Indeed, our results enable to discuss to what extent water is only "borrowed" to be used for snowmaking. This term is often mentioned by snowmaking promoters, and condemned by opponents, who consider that water withdrawn for direct or indirect (through filling reservoirs) use for snowmaking is lost. Besides the 10% evaporative water loss during the process of ice droplet solidification into ice spheres which forms the basis of snowmaking (Eisel et al., 1988, 1990; Spandre et al., 2016a; Grünewald and Wolfsperger, 2019), our results indicate that, in the situation prevailing in the Northern French Alps, there is little to no further loss of water through snowmaking. This situation is in contrast with the substantial water loss associated with irrigation, in agriculture (Boulet et al., 2020), to which snowmaking has been compared (Steiger et al., 2019). Water storage for agriculture is often questioned, in particular due to the fact that strong evaporation from the reservoirs can occur, leading to direct loss of stored water (Rodrigues et al., 2020), and that water used for irrigation is lost through the evapotranspiration of the agricultural plants and soils. Due to colder temperature prevailing at the location where water reservoirs used for snowmaking

are built, it is likely that the evaporation rate from such reservoirs is much smaller than from lower elevation agricultural reservoirs. Nevertheless, this needs to be assessed in detail. Through these two aspects (lack of "evapotranspiration" from the snowpack, and different, probably lower evaporation rate from water storage reservoirs), it appears that snowmaking differs markedly from water use for agriculture irrigation in lowlands, thus this comparison may be partially misleading from a water management perspective. To a large extent, it appears from our study that the effect of snowmaking on the hydrological regime of the catchments affected by the presence of ski resort is rather neutral at the annual scale, and mostly operates at the seasonal scale, by storing water masses temporarily, on the order of a small fraction of the water at the scale of the catchments (typically less than 1% to 2%), either in reservoirs prior to snowmaking, or in the form of snow itself through snowmaking. In terms of water quantities, snow grooming and snowmaking mainly leads to shifting the water cycle in time (at the seasonal scale) and space (through water transfers in some cases). Such shifts can have consequences for water resources management at the local scale, especially in cases where (i) wintertime abstractions in river are used for other uses of water, because snow grooming tends to lower the river discharge in wintertime and (ii) water abstractions for snowmaking take place at times and locations where conflicts for water use can be encountered. Tools such as those developed in this study could be used, in the future, to better assess and manage such situations at the local scale.

This study is consistent with the conclusions of IPCC (2019): "Integrated water management approaches across multiple scales can be effective at addressing impacts and leveraging opportunities from cryosphere changes in high mountain areas. These approaches also support water resource management through the development and optimization of multi-purpose storage and release of water from reservoirs (medium confidence), with consideration of potentially negative impacts to ecosystems and communities." Indeed, we highlight here that this study does not address all the issues related to the impacts of ski resorts operations and snowmaking on the mountainous environments. The influence on the water cycle, quantified in this study, is only addressed in terms of the effects of snow grooming and snowmaking (hence, not accounting for other water uses related to tourism activities, including housing), and only in terms of water quantity but not water quality. Also, water storage for snowmaking is only considered here in terms of water amounts and the related alteration of the water cycle, but does not address other environmental impacts on biodiversity and mountain landscape related to the construction and use of mountain water reservoirs and surrounding ski resorts. Implications relevant to the positioning of snowmaking and ski tourism within the future path development of mountain tourism and broader mountain economies (Steiger et al., 2022; Berard-Chenu et al., in press; Scott et al., 2022; François et al., 2023) are beyond the scope of this particular study, which solely focuses on downstream hydrological impacts.

### 5.3 Future climate

Our analysis is also based on future climate change simulations. The results indicate that the impact of snowmaking on the hydrological cycle at the local scale will moderately increase in a warmer climate, although the impact of climate change on the hydrological balance of the related catchment (in particular, earlier and smaller snowmelt peak) will be of a larger magnitude. This method, pending the use of a larger set of climate change simulations to better quantify the uncertainties at play, can be used to provide quantitative information combining the influence of (i) grooming and snowmaking and (ii) climate change on

the local water cycle in mountain catchments within which ski resorts are located. This will require the use of a full ensemble of climate change projections in order to provide quantitative assessments of the rate of change, along with the uncertainties at play (Verfaillie et al., 2018; Gerbaux et al., 2020). Furthermore, using ensembles of climate change simulations will enable to

compute the time evolution of water resources in the ski resorts under consideration, for future climate conditions. Simulations carried out in this work were performed using current snow management rules and technical configurations, which may evolve in the future, due to potential changes in the technologies used for snow management (snowmaking equipment, grooming techniques etc.) but also changes in the operational practices implemented in ski resorts, including those due to future changes in skier demand and other future changes in the local economies and the future role of ski tourism.

## 6    Conclusions

This study introduces a new framework for analyzing the impact of snow management on the hydrological regime at the scale of individual ski resorts. The method was implemented and demonstrated results for a case study of the La Plagne ski resort. Its generic formulation makes it possible to be replicated for other ski resorts and related catchments. The results of the case study at La Plagne shows a substantial influence of the ski resorts snow management operations at the local scale on the seasonality

of runoff amounts, especially due to grooming in winter. The effects of snowmaking on runoff generation occurs mainly at the time of snowmelt, and its magnitude depends on the geographical scale of aggregation, up to about 1% to 2% under current climate for the catchments under consideration.

The study provide hints into the impact of future climate change on the hydrological alteration due to grooming and snowmaking. For the catchments under consideration, the amount of snowmaking, under current snowmaking practices and tech-

nology, increases in a warmer climate, consistent with previous studies (Spandre et al., 2019). However, at the catchment scale, the main influence of climate change on the hydrological regime proceeds through the well documented earlier and smaller (mostly natural) snowmelt peak (Hock et al., 2019), over which the influence of grooming and snowmaking overimposes a secondary modulation. Thus, under current and future climate conditions, grooming and snowmaking appear to have a rather limited impact on total runoff at the catchment scale. The main technical issue raised by the use of snowmaking in a warmer

climate is the ability to have sufficient water available for snowmaking operations (e.g. Gerbaux et al., 2020).

Although our study provides quantitative estimates of the impact of grooming and snowmaking on the hydrological regime of mountain catchments influenced by the presence of ski resorts using a modelling framework, it also reveals the complexity of using in-situ hydrological observations in such contexts. Often, the data do not exist, or are too incomplete to provide a consistent picture, based on observations, of the hydrological context. This is a clear area of potential improvement for better

integrated water management in mountain areas, under current and future climate conditions (Hock et al., 2019), bringing together integrated modelling approaches with a broader range of in-situ observations. We hope and expect that forthcoming studies will provide additional insights on the complex social-ecological challenges faced by mountain areas under global change.

*Code availability.* The Crocus snow cover model used for this work was developed inside the open-source SURFEX project (http://www.umr-cnrm.fr/surfex/, accessed 25 Septembre 2023)

*Data availability.* Data used in this article are available from the authors upon reasonable request.

*Author contributions.* SM, HF and ES designed the research; MR, HF, ES, LC, FB and EL produced the data; MR, HF and ES produced the figures and the table with support from co-authors; all authors contributed to the analysis and interpretation of the results; SM wrote the paper, using feedback from all co-authors.

*Competing interests.* The authors declare no competing interests.

*Acknowledgements.* The project conducive to this manuscript has received funding from Compagnie Des Alpes and the European Union's Horizon 2020 research and innovation programme (Grant agreement no. 730203). We thank three anonymous reviewers, and the topical Editor Dr Elena Toth, for insightful comments and suggestions, which lead to improving this manuscript through the editorial process. CNRM/CEN and LESSEM belong to Labex OSUG@2020.

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
