# Peer review of "A model study of the local alteration of the hydrological cycle downstream of a ski resort due to grooming and snowmaking"

_EGUsphere, 2022_

## Author Response (AR1)

Dear Editor,

We are grateful to the three reviewers for the thorough and insightful comments and suggestions on our manuscript.

We have addressed all individual comments and are pleased to provide a revised manuscript, consistent with the suggestions and comments from the reviewers and our responses.

Note that we have indicated where the changes were implemented using the line numbers referring to the manuscript highlighting track-changes – note that these are not the same line numbers as in the revised manuscript where changes are not apparent. This information is higlighted in blue color in the reply to reviewers below.

We hope that our manuscript will be deemed acceptable for publication and look forward to hearing back from the editorial process.

Yours sincerely,

Samuel Morin, on behalf of the author team

Responses to Reviewer #1

We thank Reviewer #1 for his/her very useful comments, which we have taken into account to produce a revised version of our manuscript. We provide below a point-by-point reply to the comments and suggestions made along with corresponding changes to the manuscript. We hope that the revised manuscript, which, in our opinion, is improved over the original submission, will be found satisfactory for acceptance and publication in *Hydrology and Earth System Science*. Comments and suggestions are in bold and responses are in light font.

**General comments:**

**The paper addresses a relevant scientific question which is within the scope of HESS. It is attempted to quantify the effect of grooming and technical snow production on the water balance components in a Northern French skiing area by means of a combined physically based modelling experiment. In this comparably narrow scientific field the study is probably the first of its type and it does present novel ideas and data.**

**In general, this is a very valuable and novel contribution in this field. It could however profit from a sound explanation of the choice of the presented methodology.**

**The original approach of the coupled models has obviously been developed for large scale applications, but is applied here at the local scale of a single ski resort. This resulting scale gap requires several regionalization steps and assumptions which might be the cause for manyfold uncertainties. A sound argumentation should be presented why concepts are chosen like „gravitational envelopes", „Ski resorts Representative Units" or „SAFRAN altitude bands …" (and adopted from the Vanoise massif, some distance away). The spatial units are „intersected" by the ski pistes with or without their snowmaking equipment (requiring water fluxes scaling afterwards), and/or „crossed" by the slopes. Even though the original literature where these concepts are described is presented, it mostly remains unclear why the given set of methods is appropriate, and why not a method is applied which uses local measurements and reproduces the water fluxes at the local scale.**

We thank the reviewer for this positive appreciation and these criticisms, which we address in the revised manuscript, to better introduce the methods, with their benefits and limitations. This study is mostly based on modelling, which we better reflect in the proposed revised title of the manuscript entitled : "A model study of the local alteration of the hydrological cycle downstream of a ski resort due to grooming and snowmaking".

**The results indicate that the hydrological effect of grooming/snowmaking is small. To which degree are these results caused by uncertainties of the CROCUS simulations? This question arises since for the latter a set of assumptions is applied which significantly might affect the magnitude and timing of the computed water fluxes (mainly with regard to snowmaking practice, initial water loss and available water amount). Another source of uncertainty is probably the model forcing at the SRU scale, since the chosen method does not account for the conditions at the location of the snow guns and lances. Ski resorts applying technical snowmaking usually**

**monitor and save the available water storage volume and fluxes used for the snow production, so this data should be available.**

Indeed, some of the data for snow production is available and used in the study. We describe uncertainties related to modelling choices in more details in the revised manuscript. Note that we refer to studies, which explicitly assessed some of the key uncertainties related to modelling snow on ski pistes, such as Ebner *et al.* (2021). This is reformulated for better clarity in the revised manuscript (Page 24, lines 560 to 569). Also, while the study is mainly based on modelling, we have added further comparison with observations, including on discharge measurements in the Frasses catchment (Page 16, lines 372 to 388) and the inclusion of a new figure (Figure 6, Page 17).

**Two catchments are introduced, one of which is ungauged and requires a spatial transfer method, and generally „information on hydrology is rather sparse". Hence, several simplifying hypotheses are formulated. Is it possile to evaluate the effect of these hypotheses at the local scale where measurements are available? If the approach remains „purely empirical": could it be replaced by a much simpler, but easier to understand estimation?**

Our study is mostly based on modelling, and is better introduced as such in the revised manuscript. We provide further information on the uncertainties related to the methods used in this study in the revised manuscript (Page 24, lines 560 to 569).

**The overall presentation of the paper is well structured and the authors give proper credits to related work. The abstract would benefit from a more complete presentation of the most important results.**

The revised abstract includes more quantitative results, following the suggestion from all reviewers (Page 1, lines 21 to 31).

**The choice and number of references is adequate.**

**Specific comments:**

**- some terms used are not very common in hydrology (e.g. „disturbances", „disruption", „behaviour", „alteration"). I recommend to change the title accordingly, and also choose other terms in the text**

The revised title reads "A model study of the local alteration of the hydrological cycle downstream of a ski resort due to grooming and snowmaking".

The terms "alteration" and "altered" is used throughout the text in accordance with those used by the USGS (e.g. https://www.epa.gov/caddis-vol2/flow-alteration) and the European Environmental Agency (https://www.eea.europa.eu/publications/regional-water-report).

**- the analysis of the climate change effects is of limited explanatory power, since only climate is considered. However, many other influencing factors - hard to predict, though - will change and develop in parallel to the climate. It is not so clear if the conciseness of the paper profits from this section**

Challenges related to ski resorts operations and related pressures on the local environments, in particular water resources, under future climate change, is a prominent and controversial issue in mountainous environments. Here we provide a first-of-its-kind assessment of the alteration of the local water cycle related to snow grooming and snowmaking under future climate change, which we expect will motivate further work on the matter. We have clarified, while keeping it short, the description of these results (Page 20, line 493 to Page 24, line 543) and introduced a revised Figure 9 (Page 33).

**- „water reaching the soil": to which degree is this a suitable hydrological variable for hydrological change? It can affect streamflow amount and timing at the catchment outlet in very different ways, depending on the hydrological characteristics of the catchment. Maybe one could still add a simple consideration to relate water amounts to streamflow regime, as announced in the text ?**

This variable combines rainfall on snow-free ground and snow melt (representing all of the water flowing from the snowpack at its bottom), and is thus directly related to the water entering the water cycle at the local scale. We discuss and clarify its suitability in the revised manuscript, including through a comparison of the results obtained using this output variable from the model results with hydrological observations. See e.g. the revised discussion on the limitations of this approach (Page 24, lines 560 to Page 25, line 578).

**- is mechanical stress caused by the skiers and its effect on the snow surface considered in the simulations?**

The simulations take into account snow grooming and snowmaking. The direct effect of skiers is not represented in the model but the grooming component represents the densification related to ski resorts operations and the effect of daily application of grooming on ski pistes. Previous research has shown that the density profile of the snowpack on ski pistes (including skiers) is adequately represented by model results (e.g. Spandre et al., 2016).

**- does SAFRAN provide humidity? How is wet bulb temperature derived?**

The SAFRAN near surface atmospheric reanalysis includes relative humidity, which is combined Crocus to infer the wet bulb temperature used for the snowmaking threshold. This is described in Spandre *et al.* (2016).

**- technical snow is rather different than natural snow. On the slopes with snowmaking, a mixture of the two develops over the season with varying composition and hence changing physical properties at the surface. Does CROCUS account for that?**

Snow on ski pistes is indeed a mixture of machine-made snow and natural snow. The representation of grooming includes a mixing of layers (effect of the tiller), leading to an homogeneous mixture of natural and machine-made snow, as described in Spandre et al. (2016) which also provides a comparison between simulated and observation snowpack profile on ski pistes.

**Technical corrections:**

**- figures should be larger**

We have increased the size of the legends of the figures, which will increase their readability. We are open for further suggestions from the HESS copy-editors to further enhance the figures depending on the final formatting (simple or double-column).

**- the English language could profit from correction by a native speaker (mainly: uses of articles, and singular/plural)**

We have thoroughly checked the language of the revised manuscript, to the best possible extent.

**- better explain the basic functioning of HydroDem explicitly, rather than referencing another software (TauDem)**

The HydroDem software is presented in more detail in the revised manuscript (Page 9, line 215 to Page 10, line 234).

**Congratulation to this work and valuable contribution! I hope my comments support the further improvement of the manuscript.**

We are grateful for the useful comments and suggestions, which have clearly helped to prepare a better revised manuscript.

Reviewer #2

We thank Reviewer #2 for his/her very useful comments, which we have taken into account to produce a revised version of our manuscript. We provide below a point-by-point reply to the comments and suggestions made along with corresponding changes to the manuscript. We hope that the revised manuscript, which, in our opinion, is improved over the original submission, will be found satisfactory for acceptance and publication in *Hydrology and Earth System Science*. Comments and suggestions are in bold and responses are in light font.

**Morin et al. present a very interesting study on the hydrological implications of snow grooming and snowmaking in the French Alps. Despite being a frequent practice, such studies are rare and our hydrological understanding is limited. To achieve this goal, they implemented a snow model (Crocus) for a ski resort and compare simulations with or without grooming and snowmaking. They found that the influence of such techniques on the hydrological cycle is significant at slope scale, while it expectedly decreases when looking at larger scales.**

**Overall, the study is interesting and novel and I am in favor of publication. At the same time, it presents some points that should be better discussed, especially for the fact that the paper aims to discuss hydrological implications but no full-scale hydrological model was implemented. Authors openly discuss this, but at the same time some conclusions related to the impact of ski-resort procedures on the hydrological cycle should be refined in this sense (see below).**

We thank the reviewer for his/her general appreciation of our work and for suggestions which are helpful in better conveying its results and their discussions, which we have taken into account for preparing a revised manuscript.

**SPECIFIC COMMENTS:**

**- The abstract reads quite qualitative at places, and I would recommend being more specific and quantitative. For example: "several km2" (please report the exact extent and location); "visible impact" (please quantify this); "few percent" & "additional snowmelt amount" (same, please quantify this);**

The revised abstract takes into account more quantitative results, although we make it clear that the quantitative results are only applicable within the scope of this particular study in this specific location. (Page 1, lines 21 to 31).

**- Introduction: I missed specific research questions and/or hypotheses that authors want to explore. The overall goal of the research is clear, but the lack of specific research questions makes the flow of the paper a bit difficult to follow. Also, please consider breaking this section into paragraphs for readability;**

The main research question may sound too "easy" to formulate, but in fact this manuscript aims at characterizing to what extent the presence of a ski resort alters the local hydrological cycle through grooming and snowmaking. The introduction of the revised manuscript is rephrased to better identify this clear question motivating this study. See in particular Page 2, lines 52 to 54.

**- Line 82: could you please elaborate a bit more on these "deviations"?**

"Deviations" referred to here correspond to various settings of the snowmaking equipment (flow rate, temperature threshold, etc.). This is described in the revised manuscript (Page 6, lines 138 to 139).

**- Section 2.3: the concepts of SRUs and gravitational envelopes are interesting, but also a bit hard to grasp for readers like me that are not familiar with this particular field of study. How do SRUs exactly differ from hydrological response units? My understanding is that they are found intersecting elevation and aspect of ski slopes with location of snowmaking equipment, but this was a bit too concise. Same for the gravitational envelope concept. Perhaps a schematic might help here?**

HRU and SRU are indeed very similar in terms of concept, and the "Ski resorts Representative Units" (SRU) have in fact derived from the well-known HRU concept. This has been described in detail in previous publications (e.g. Hanzer et al., 2020, Ebner et al., 2021) and the revised manuscript is improved for better clarity regarding these terms and concepts, while remaining as concise as possible (Page 8, lines 186 to 195).

**- Section 2.4: could you elaborate a bit more on the adjustment method used (is it a bias correction?) and perhaps spend a few words on the S2M reanalysis?**

The revised manuscript elaborates more about the S2M reanalysis and bias correction method (ADAMONT) used to adjust EURO-CORDEX climate projections, in a dedicated subsection of the Methods section (section 3.5, Page 300 to 301).

**-Figure 2: please consider adding a small panel with the location of this resort in France;**

This figure is updated accordingly in the revised manuscript (Figure 2, Page 5).

**- Results: please consider summarizing section titles, which are quite long at the moment**

Subtitles are shortened in the revised manuscript, while retaining the information on the scale of the analysis (3.1 Hydrological alteration at 1800 m elevation, 3.2 Hydrological alteration over the gravitational envelope of the ski resort, 3.3 Hydrological alteration at the catchment scale).

**- Section 3.3 (comment #1): while the discussion of snow simulations was very clear, relevant, and interesting, I got a bit lost in this section about the catchment scale. My main issue was that I didn't quite understand how authors combined the different sources of information, and how these sources of information interacted with each other to solve the overall water balance and so derive Figures 6 and 7. I would suggest the inclusion of a water balance equation where all terms are reported, so to understand how the impact of sky resort practice was quantified.**

Hypotheses are presented in detail for a better understanding of the results, see Page 17 line 414 to Page 18, line 435).

**- Section 3.3 (comment #2): my understanding is that the various sources of data refer to significantly different periods, sometimes with no overlap (e.g., water resources data date back to the 40s-60s). Perhaps I am missing something here, but I would include a discussion on comparability of climate across the various periods, since water use and consumption can significantly change based on precipitation and temperature patterns. Also, are "water resources" streamflow measurements in the end?**

Only first-order estimates of the water footprints developed along the river system can be expected due to limited information on the snowmaking process, water abstraction and consumption, and to the fact that observation data cover discontinuous and non concomitant time periods.

For example, the water balance is biased by non-concomitant data periods (dating back to the 40s and 60s for water resources and after 2007 for abstractions), whereas the demand for water for snowmaking and reservoir management is correlated with the meteorological and hydrological conditions of the year.

Water resources refer to river flows, which are considered representative of natural water availability.

This clarified in the revised manuscript, see e.g. Page 18, lines 442 to 445.

**- Line 280: what do you mean with "hides heterogeneity" here?**

River flow regimes and their alteration due to human influences differ from valley to valley.

**- Section 4.2: I agree with authors that evaporation losses during winter are small, and so snowmaking consumption is expected to be minimal in that regard. However, shifting snowmelt peak by weeks to months (as reported in Figure 5) implies that the bulk of snowmelt is mobilized at a time when transpiration by plants is much higher than in winter. Also infiltration patterns are very different from early spring to early summer, depending on soil thermal and moisture conditions. In this regard, this in-depth discussion on water management implications would benefit from some forms of hydrological modeling, or at least a quantification of the other components of the water balance (evapotranspiration and storage).**

We agree that a more detailed hydrological framework would make it possible to assess indirect effects of snow management, such as the influence of the shift in snowmelt timing on evapotranspiration. We discuss this in more details in the revised manuscript (Page 24, lines 561 to 565).

Reviewer #3

We thank Reviewer #3 for his/her very useful comments, which we have taken into account to produce a revised version of our manuscript. We provide below a point-by-point reply to the comments and suggestions made along with corresponding changes to the manuscript. We hope that the revised manuscript, which, in our opinion, is improved over the original submission, will be found satisfactory for acceptance and publication in *Hydrology and Earth System Science*. Comments and suggestions are in bold and responses are in light font.

**The manuscript presents an assessment of the influence of snow management on water resources at a ski resort with heavy snow making. It is a very interesting topic, and while being very niche, of high interest to the recreation and ski resort management community. It is interesting to see how this study brings in physically-based snow simulations with water management assessments to find some insights on a relevant issue to the local community – it is novel, and innovative while being oriented towards water management issues. I overall enjoyed the manuscript and found it straightforward and interesting. I appreciated the clear discussion of the implication and limitations.**

We thank the Reviewer for his/her positive appreciation of the results and the manuscript.

**Overall, I would have liked more information about the snow production numbers – how often and how much ends up being made. This seems like an important aspect of the discussion, but it was glossed over. I would also have liked more context on the methods – why certain methods were chosen, and especially how the water supply and demands was classified in the hydrological assessment.**

The revised manuscript includes more information on the matter. Information about the total water amount for snowmaking is provided in Table 1, Page 20. The new Figure 9, page 33, also provides further information about the water demand for snowmaking and its comparison to other terms of the hydrological balance. This is discussed in the text, Page 25, line 604 to Page 26, line 626.

**My main concern with the manuscript is the lack of model evaluation. The proposed results could be completely different from how snowpacks evolve in the area. Could the snow evolution presented in Fig4 be compared with satellite imagery of actual snow cover in the basin? Could the water supply (or streamflow) in Fig 6-7a be compared with the limited streamflow measured in the basin?**

The study is mainly a modelling study and this is now better reflected in the very title of the study, which in its revised form is entitled "A model study of local alteration of the hydrological cycle due to grooming and snowmaking downstream of a ski resort". However, the modelling tool used in this study was extensively evaluated in previous studies, including snowpack modelling on ski pistes, see e.g. Ebner et al. (2021). In terms of hydrological observations we recognize that the observations are heterogenous and incomplete, but we use the available information as much as possible to infer relevant and meaningful information through the combination of all available data sources. This is better described in the revised manuscript, which also includes a complementary comparison between

simulations and hydrological observations for the Frasses catchment (Page 16, lines 372 to 388), with the addition of an additional figure (Figure 6, Page 17).

**I also suggest an examination of the language to make sure the sentence structure is clear and without too many clauses. Some sentences are long, with many commas and additional information. It makes them hard to follow. I also suggest the authors try to use precise language throughout the manuscript ("several, a few, many, some, etc, … "could be replaced by the actual numbers or words). I also have many small comments to clarify some language used or some suggestions on the figures.**

The revised manuscript was thoroughly checked, clarified and improved for such issues.

**In-line comments:**

**L10 – Can you give the number instead of "several"?**

The revised manuscript includes more quantitative figures.

**L14 -Same thing – instead of saying "few percent" could you give the actual number?**

The revised abstract includes more quantitative results, however we clearly indicate that the results obtained are specific to this study and this location (Page 2, lines 21 to 31).

**L15 – This might be a tricky one, but in North America, we don't call them piste – we refer to them as ski runs. Maybe something like a skiable area would be less regionally-dependant?**

This is a good point, which occasionally triggers discussion, and sometimes inconsistency in the literature. The revised manuscript uses the term "ski pistes" throughout, which is less ambiguous than many other terms (skiable areas, ski slopes etc.) which sometimes could be understood as including off-piste terrain.

**L80 – Wouldn't wind fit in the atmospheric conditions? What do you mean by "etc"?**

The sentence is reformulated in the revised manuscript, for better clarity (Page 5, line 135 to Page 6, line 137).

**L123 – Can you add some information on this? What is the Vanoise massif – does this refer to the mountain range where La Plagne is, or the closest location with the forcings? What makes it representative or adapted or beneficial for your application?**

The "Vanoise" massif refers to the corresponding area as part of the geographical partitioning of the French Alps used in the SAFRAN-Nivo reanalysis, used to drive the Crocus snow cover model (S2M reanalysis, see Vernay et al., 2022). The S2M reanalysis assumes that meteorological and snow conditions are constant within each massif, and only vary with elevation. The La Plagne ski resort is included in the Vanoise massif, and it is typical for many studies of the meteorological and snow conditions in the French Alps to use the S2M reanalysis (Vernay et al., 2022).

**Figure 1: If snowfall falls on the ground, it should go into the snowpack? I don't quite understand how you can have a snowfall arrow on the snow-free ground. If it melts instantaneously, it would still be liquid water reaching the ground?**

If snow falls on a bare ground, it indeed contributes to creating a snowpack, which may eventually melt, immediately or not, and contribute to the total water reaching the ground through snowmelt. The two cases on the Figure are meant to differentiate between the situation of liquid precipitation (rain) which needs to be taken into account directly only in the case where it falls on snow free ground, otherwise it is intercepted by the snowpack and may be incorporated, through the percolation processes, together with snowmelt (= all the water flowing at the bottom of the snowpack).

**Also, do you ignore completely anything related to forest interception? The snowfall on the piste and off-piste will be quite different. This will not apply to comparing groomed and groomed+snowmaking, but it might apply when considering the sub-basin assessment, or the gravitational envelope assessment.**

In this modelling study we do not take into account forest interception, and compare simulations with and without the presence of snow management on ski pistes (grooming and snowmaking), everything else being equal. We clarify this assumption in the revised manuscript (Page 7 lines 161 to 163).

**L133 – ref for the DEM DB-alti?**

The reference is added in the revised manuscript (Page 9, line 205).

**L136 – Why do you give a reference to taudem, but not to hydrodem? I am also not sure if comparing HydroDEM and taudem and mentioning that tauDEM is more well-known is relevant – arguably, different science communities will know one or the other better. You could mention if one is more recent than the other, or if one has different benefits or flaws, but I don't think "well-known" is a solid argument. Also, you write it HydrodDem and HydroDEM. Please be consistent.**

The HydroDem software is presented in more detail in the revised manuscript (Page 9, line 215 to Page 10, line 234.)

**L141 – Can you specify the years instead of mentioning "previous years"**

This is clarified in the revised manuscript (Page 12, lines 296 to 298).

**L145 – I understand that most of the information about the adjustment to the local scale is done like in Verfaillie paper, but could you add 1-2 sentences summarizing it here? It is an important step. Also, what is the spatial and temporal resolution of the GCM/RCM product? And, you mention that you take the result for RCP8.5 for 2043-2057, but then in L153, you mention that you use the values for the adjustment for 2090-2099? Could you please add some more information about these products and downscaling to help understand this manuscript?**

We have clarified that the ADAMONT method is a quantile mapping adjustment technique (Page 12, line 302 to 303). It uses daily RCM model output as an input for the statistical

adjustment. While in the manuscript we indeed focus on the periods 2043-2057 and 2083-2097, see e.g. on Figure 9), we provide context for this sole GCM/RCM pair based on an analysis contrasting with other GCM/RCM pairs in Monteiro et al. (2022), which focused on 2090-2099.

**L154 – I suggest removing "pending further investigation using a genuine ensemble for climate model simulations" and L149 "although this study does not address…". I appreciate the honesty about the limitations of the products and study, but I suggest moving these mentions of limitations to a discussion section.**

Such caveats and limitations are important to note and are briefly provided here in the introduction of the methodological approach in section 3.5 Page 12 lines 309 to 310.

**L158 – Could you give this value in km2 or m2?**

The value is now given in km².

**L161 – Can you add a sentence about what a lance is?**

A lance is a type of snowmaking equipment, using pressurized water rather than a fan-like snowgun. French ski resorts are equipped to a great extent with such equipment. This is introduced with more details in the revised manuscript (Page 4, lines 99 to 100).

**Fig 2 – The colours for the elevation bands are hard to distinguish, especially with the hillside applied. If it is important to see the 300m elevation bands, I suggest having more distinctive colours. Otherwise, you could use a continuous gradual colour gradient map instead of a classification with 9 types. These maps with the Bonnegarde catchment come before any explanation of the Bonnegarde basin in the text. I suggest either moving the figure down or mentioning in the caption that Bonnegarde is the main studied sub-basin. I also don't think you need to mention that you made the map. I also find Fig 2 and 3 somewhat redundant. Could you also put the Frasses basin on Fig 2a? or maybe merge Fig 2 and 3?**

We change the colored bands in the revised manuscript, in order to make them easier to distinguish. We added the "made by" mention to avoid any confusion, i.e. the map is not related to the resort manager, and it is not responsible for its content.

Fig 2. and 3. show different information levels, the basin in the context of the full resort on the one hand and the details of the studied basin on the second one. Merging the two maps would provide too much of information at a time and we prefer keeping both figures for the sake of readability.

**L177 (point 1.) – Could you add this point on Fig 3?**

This is the red point in the revised map, named "gauging station".

**L185- Could you remove the "… "- either mention "such as industry, energy production irrigation) or give the full list.**

The sentence is modified in the revised manuscript (Page 11, line 287).

**Fig 3 – What are the small grey lines outside of the basin? The two small subbasin maps on the right (could you add a-b-c labels to your panels?) are not providing much new information and could maybe be removed, and then Fig 2-3 could be joined. For the easiness of map reading, could the basin outlines lines be made thicker? And the points and text? As you mention in the text, the local of BNPE cannot be put on the map exactly, but you have it on the map, which is misleading.**

The grey lines refer to rivers outside of the catchments studied in the study. Other modifications have been done. We have decided to keep the withdrawal points extracted from the BNPE database and, to avoid confusion, we mention in the legend that their location is approximate. However, as indicated above, we do not merge the Figures 2 and 3 but clarify and simplify their content in the revised manuscript.

**L192 - You don't have a section about model evaluation?**

Individual model components were evaluated in previous publications (e.g. Spandre et al., 2016, Vernay et al., 2022 and Ebner et al., 2021). We add some further comparisons between simulation and observation data in terms of river flow in the Frasses catchment, in the revised manuscript (Page 16, lines 372 to 388 and Page 17, Figure 6).

**L196 – You run the simulations for a much longer period – why do you show only 2019-2020 as an example? Have you considered doing the average of the period? Or explain why you only show 1 year of results.**

Figures 4 and 5 are mainly intended to provide examples of the model output, in order to illustrate the main effect of grooming and snowmaking on river flow at the point and catchment scale. Figure 8 (now Figure 9 in the revised manuscript) provides a multi-annual perspective on the results, under past and future climate conditions. It has been considerably expanded, compared to the discussion manuscript.

**L200 – I don't quite follow how a higher thermal conductivity results in cooling – Here are you specifically talking about radiative cooling at night? Because then, wouldn't the transfer of energy in the day, causing faster melt, also occur? Do you mean that the snowpack loses its insulation capacity over the ground? Also, if this is background knowledge that has been studied in previous literature, this description of how grooming influences sow behaviour should be in the introduction, and then could be further discussed in the discussion.**

This effect was indeed described and discussed in Spandre et al. (2016). Indeed, a groomed, hence denser snowpack, loses much of its insulation capacity over the ground, leading to more efficient energy loss especially at night. While this does not need to be mentioned in the introduction, because it is not central to the scientific question addressed by this study, we refer to this process here together with the description of the Figure 3. We clarify that this is previous knowledge in the revised manuscript (Page 12, line 320).

**L207 – This also means that you neglect any kind of snow interception and sublimation by trees or blowing snow sublimation during a blizzard. I assume that a lot of the off-piste area is forested (as in Fig 2 ski area map) in the study area – how**

**do you represent forest hydrology in the model? And how does it influence your results? You don't mention the forest within the manuscript.**

Indeed, we neglect the potential role of forests and focus on differences between the situation with and without grooming and snowmaking on open areas where ski pistes are located. This choice is better introduced in the revised manuscript (Page 7, lines 161 to 163).

**Section 3.4. – Could you provide a number relating to how much snowmaking occurs over the different years? in the methodology, you mention it could occur throughout the winter, but in the figure, it only occurs at the beginning of the season. Is this the case in all the simulations? Are the values for the 2019-2020 representative?**

Most of the snowmaking indeed occurs in the beginning of the season, corresponding to operational practices (see Spandre et al., 2016b). The figures providing all of the components of the water cycle (relevant to this study), namely Figures 7 and 8 of the revised manuscript, show the seasonal distribution of snow production.

**Fig 4 – same as the previous figure – make the text and line width larger so the figure is easier to read. In the caption, you say "La Plagne ski resorts" – is there more than 1 resort? (resort vs resorts?). You are also missing the word "year" for the hydrological year 2019-2020. Have you considered plotting the individual fluxes for each simulation instead (grouping per fluxes instead of per simulation?) I am mainly interested in how snowmelt changed between the simulations, and it is hard to see as they are on 3 different panels. The snowfall and rainfall lines are also redundant, as they don't change between the simulations. And what are the vertical grey lines?**

The text and line width are larger in the revised figure. In addition, the gray lines (marking each month) have been removed in the revised figure and the variable snowmelt is shown in the revised figure as a separate panel. Also, rainfall and snowfall have been removed from the two last panels and kept only once as they are the same for all simulations. This makes it easier to compare between the subplots, even if we did not change the overall structure of the figure. The caption has been modified and clarified based on the comments.

**L240 – and the differences between the simulations here are highly over exaggerated as you consider the runs to be completely covered in snowmaking, compared to the 40% in reality.**

Here we provide results for idealized, simplified configurations. This is highlighted in the revised manuscript. Simulations for the Bonnegarde catchment, Figure 9 (of the revised manuscript), takes explicitly into account the fractional coverage of snowmaking on ski pistes.

**L245 – Can you explain what you mean by empirical here? Do you mean it is a first-order estimate based on the limited information and modelled result?**

Exactly. The sentence has been rewritten (Page 16, lines 396 to 397).

**L251: Do you mean 'assumptions" – are these assumptions of how the basin behaves?**

Assumptions are related to water management at the catchment scale. They are needed due to lack of information.

**L258 – I don't follow – where do the 60% and 30% values come from?**

These values are derived from the snowmaking modelling (section 2.1). A water loss of 40% is considered, but no details were given on the split between evaporation losses (of the order of 10%) and losses due to the fact that some of the snow produced does not fall on the ski slopes (of the order of 30%). The text of this section is modified accordingly, in the revised manuscript (Page 18, lines 424 to 432).

**L262 – You use the hydrological estimates based on 100% of the ski runs being covered by snowmaking, but then you only apply the volume of snow made to the areas with snowmaking. This seems like a mismatch of approaches.**

This assumption is only used to distribute in space the volume of snow produced on each ski piste equipped with snowmaking in the catchments.

**L245 to 262 are details regarding the methods and should be in the methods section, not the results.**

In the revised manuscript, some sections are moved to follow a better flow of information with a clearer separation between Material and methods, Results and Discussion.

**L266 – You mention Fig 6-7 before table 1, but then table one comes first in the text. Please adjust either the text or the order of the figures.**

The layout will be revised in the published final version to ensure a relevant order of appearance of Figures 6 and 7, and Table 1 and to make the comparison of Figures 6 and 7 easier.

**L269 – Maybe add a bit more information about the results from Table 1?**

The revised manuscript refers more to the results from Table 1 (Page 19, lines 484 to Page 20, line 491).

**L277 – "for areas at high elevation" – this is confusing – what does this sentence clause refer to?**

The ski resort of La Plagne and its accommodation are located at different altitudes of the massif. It is not necessary to distinguish between the different areas given the scale of the study. "for areas at high elevation" is deleted in the revised manuscript.

**Fig 6-7: In the text, you keep presenting the result for Fig 6a-Fig7a, and Fig 6b-Fig7b, which forces the reader to go back and forth between the two figures. Could you either put the two figures side-by-side or maybe with bars side by side? Something to make the comparison easier?**

If the manuscript is accepted, the layout will be revised in the published final version to ensure a relevant order of appearance of Figures 6 and 7, and Table 1 and to make the comparison of Figures 6 and 7 easier.

**I also don't think it is okay to put the solid and liquid contribution in volume – the density between the snow and the liquid water is different enough that this comparison between snow production and grooming and the snowmaking effect (or other subpanels) is misleading. This should likely be transformed to liquid as well to take into account the difference in density.**

Water amounts expressed in volume can be ambiguous when it comes to snow. The revised manuscript mentions that Figures 6d and 7d show the equivalent volume of water for snow in the text (Page 18, lines 446 to 447).

**Fig 6-7:**

**- By water resources, do you mean streamflow? The amount of water available in the river?**

Water resources refer to river flows, which are considered representative of natural water availability.

**- The axis for e-f is missing a number.**

The two figures have been modified.

**- I am curious about your choice to start at week 31 - it splits your JJA into 2 sections and what the different colours represent (light blue, black, green, dark blue). And also, the figure is quite long, and it would be nice to have the weeks on the x-axis in more than one location to avoid having to always scroll back to the top.**

We consider in all figures, when relevant, the hydrological year starting from 1st of August of any given year and ending 31 July of the following year (see Figures 4 and 5). Unfortunately, this definition leads to a split of the summer period into two sections.

The colours on these two figures were chosen to better distinguish the different types of water use (green and blue for water supply and for grooming and snowmaking, respectively) from natural surface water availability (black). The shades of blue distinguish between the solid (S, dark blue) and liquid (L, light blue) phases of snow.

**- I find the classification of what is what in the subplot hard to follow. Fig 6-7b, you call it drinking water but in L272, when you mention the figure, you call it water supply? And for Fig 6c, you call it "abstraction for reservoir filling", but in L279, when you refer to the subplot, you call it weekly water abstraction for snowmaking. I suggest working on this section so that there is a clear link between what the subplot refers to and the values that were obtained.**

We have changed some terms for better clarity: for example, domestic use of water instead of drinking water (too specific), in order to clarify the text. The section is now better organized to make it easier to read the figures.

**L280 – If it is misleading to aggregate reservoir volume across the basin, then I suggest you don't present it as a result and instead show the variability between the reservoirs.**

The water balance is calculated at the catchment scale, so the volumes of the reservoirs filled with water resources available in the Bonnegarde catchment are aggregated for this purpose. The pattern displayed in Figure 7.c is not necessarily representative of all reservoirs located in the studied area.

**L320 -moderate? I think minimal, or very limited, might be a better description. It is a change of 0.22%...**

"Very limited" is used instead, in the revised manuscript (Page 23, line 538).

**L321 – The amount of snowmaking is bound by the model – meaning that snowmaking cannot be higher than a certain amount. Based on the criteria of the snowmaking processes (temperature, humidity and water availability), can snowmaking even occur? It might be more interesting to discuss how snowmaking abilities in the future might change. Would it need to occur later in the year due to temperature?**

The model accounts for wet bulb temperature thresholds for snowmaking, hence if there are situations, under current or future climate change, in which the wet bulb temperature is too high for snowmaking, it will not occur in simulations. Our results indicate that, at the elevations covered for this case study, even at the end of the century in a high greenhouse gas emission scenarios, wet bulb temperature values are sufficiently often lower to the snowmaking threshold so that snowmaking remains generally possible at least for some days of the winter, in terms of meteorological conditions. The impact of climate change on the ability to operate snowmaking equipment is not the core question of the present study, and is addressed in other studies (e.g. Gerbaux et al. 2020).

**L322- mm3?**

These are millions of m3.

**L325 – critical? Given the results presented here, which suggest a limited influence of snowmaking on water resources, this seems like a strong wording. "much-discussed" or "often debated" might be more accurate.**

In the revised paper this sentence was deleted.

**Figure 8: Monthly sums instead of monthly cumulative?**

The revised caption of the Figure 9 (previously 8) takes this comment into account.

**L325 – This first paragraph of the discussion does not bring any additional information. I suggest removing it and integrating this information either in the introduction or the conclusion.**

We have shortened this paragraph (Page 24, lines 546 to 554).

**L340 – You don't present results relating to basal melt – just about the snowpack as a whole. If basal melting is the main process that is suppressed, and this is a key result, then it could be presented in a figure of natural snow melt fluxes vs groomed snow melt fluxes in the first result section about the point balance.**

We added snowmelt specifically on Figure 4, to better illustrate this point. Note, however, that in the presence of a snowpack, snowmelt and total water reaching soil are similar, because total water reaching soil is the sum of snowmelt (water flowing from the bottom of the snowpack) and rainfall on snow free ground.

**L375 – Could you write this as an equation? It might be cleaner than only in words. Also, this rule of thumb could be a result. Or a key discussion point on its own? You present a number here about snow production that would have been interesting in the results section, as I mentioned in some other comments. If the main dilemma for water supply is the use of water for snow production, then these numbers should be presented further and put in context.**

Following this suggestion, we introduce the alteration of the hydrological regime in the form of an equation, in the revised manuscript, and discuss it specifically (Page 26, lines 610 to 617).

**L399 – Could you rephrase "a few %" to the actual number?**

We insert the actual number from the case study in the revised manuscript - although we insist here that the number may not apply similarly in a different context (Page 26, lines 622 to 629).

**Section 4.2 : You don't bring up here the fact that, based on your simulations, there is a difference between snowmaking and grooming for the ski runs, but there is such a small percentage of the basin, and with only a section of these being covered by snowmaking capabilities, that overall it makes a small difference – which is a key message I understood from your analysis – Fig 5b.**

In the revised manuscript, we better discuss the different influence of grooming and snowmaking (e.g. Page 27, lines 652 to 656).

**L421 – resort instead of resorts.**

Fixed in the revised manuscript.

**L427 – I feel like I missed that information! This is interesting.**

This is better reflected in the results section of the revised manuscript (Page 23, lines 513 to 522).

**LL428 – typo – at**

Fixed in the revised manuscript.

**L435 – The lack of evaluation data for your model should be a key point of your discussion, not one sentence in the conclusion. The entire manuscript is a modelling exercise that presents no comparison with reality. Which makes it very limited. I understand that by comparing groomed vs groomed+snowmaking, you avoid some of the complications linked to model evaluation, but you do still present a lot of results (snowpack evolution, water supply etc) that are not just the difference between simulations.**

This is better discussed in the revised manuscript, in addition to the fact that we now include further evaluation of the estimated discharge based on the SAFRAN/Crocus simulations in the Frasses catchment, compared to observations (Page 24, line 555 to Page 25 line 578).

**L439: I suggest removing this quotation from the IPCC and also rewording this final segment. You really focus on all the aspects of this study that you didn't consider (which could be put in the limitation section of the discussion), which makes the paper end on a sad note. While this manuscript has a lot of limitations, you tried to quantify a water supply issue faced in mountain communities that is highly discussed and has local relevance. I suggest you should focus on having worked on that instead of on all the things you did not solve – and that are, as you mention, outside the scope of this manuscript.**

The conclusion is rephrased to better reflect on the results of the study, and provide the limitations and references to the IPCC report in the dedicated limitations section of the Discussion (Page 27, lines 657 to 669).

---

## Author Response (AR2)

Toulouse, Lyon, Grenoble, October 2023

Dear Editor,

We are grateful to you and the three reviewers for the work on the revised manuscript, and your assessment that the manuscript could be accepted subject to technical corrections.

We have taken into account the last round of suggestions, mostly by Reviewer #1, see below.

Regarding the general comments, we agree that it is now quite late into the editorial process to shorten the manuscript.

We also hope that this manuscript will trigger further investigations, by other teams as well as by us, to further refine the modelling tools used and also to apply it other contexts so as to lead to more generalizable conclusions.

Regarding the quality of the english text : we have checked again the text, and we note that the Copernicus typesetting process includes an efficient english language editing step, which will ensure that the quality of the final text is further improved.

Yours sincerely,

Samuel Morin, on behalf of the author team

Detailed comments and suggestions by the reviewers

Reviewer #1

**I still would recommend to better formulate something like „Simulated hydrological effects of grooming and snowmaking in ski resorts on the water balance" and avoid the „alteration" term (even though it is used by USGS or EEA; there the context is a different one). „Alteration" suggests that some significant alteration effectively occurs, but the result is that the latter is almost none.**

The revised title was « A model study of the local alteration of the hydrological cycle downstream of a ski resort due to grooming and snowmaking », and Reviewer #1 proposes a different one : « Simulated hydrological effects of grooming and snowmaking in ski resorts on the water balance »**.** We have taken this suggestion into account and propose a newly revised title as « Simulated hydrological effects of grooming and snowmaking in a ski resort on the local water balance »

**The reference from HydroDem to TauDem is still there (l 215); I do still recommend to explain what HydroDem does, and not to refer from one to another software.**

We have removed the reference to TauDem.

**In the new text parts are still several mistakes. This needs careful correction prior to submission.**

We have checked again the text, and as indicated above, the manuscript will benefit from the english language check performed during the typesetting process by the Copernicus staff.

**The manuscript in its present form is very long. I wonder if it would profit from shortening here and there where possible?**

We agree that the manuscript is quite long, but, as indicated by the Editor, it is now quite late in the editorial process to perform major adjustments of the text content and length. Hence we keep it as it is and expect that the combination of text and figures in the final, formatted article, will effectively convey the main messages and results of this work.

Reviewer #2

N/A

Reviewer #3

**My only comment is to check the use of "piste" vs "slope" within the manuscript. "Ski slopes" is still used in some section instead of "ski pistes".**

We have checked again the text and adjusted the text where is was needed, in response to this suggestion.